# DART-Eval: A Comprehensive DNA Language Model Evaluation Benchmark on Regulatory DNA

**Aman Patel**[1*]    **Arpita Singhal**[1*]    **Austin Wang**[1*]
**Anusri Pampari**[1]    **Maya Kasowski**[2,3]    **Anshul Kundaje**[1,2]
[*]Equal contribution (authors ordered alphabetically)
[1]Department of Computer Science, School of Engineering, Stanford University
[2]Department of Genetics, School of Medicine, Stanford University
[3]Department of Pathology, School of Medicine, Stanford University
{patelas,arpitas,atwang,anusri,kasowski,akundaje}@stanford.edu

## Abstract

Recent advances in self-supervised models for natural language, vision, and protein sequences have catalyzed the development of genomic DNA language models (DNALMs). These models aim to learn generalizable representations of diverse DNA elements, potentially enabling various downstream genomic prediction, interpretation and design tasks. However, existing benchmarks do not adequately assess the capabilities of DNALMs on an important class of non-coding DNA elements critical for regulating gene activity. Here, we introduce **DART-Eval**, a suite of representative benchmarks focused on regulatory DNA to evaluate performance of DNALMs across zero-shot, probed, and fine-tuned settings against contemporary *ab initio* models as baselines. DART-Eval addresses biologically relevant tasks including sequence motif discovery, cell-type specific regulatory activity prediction, and counterfactual prediction of regulatory genetic variants. Our systematic evaluations reveal that current annotation-agnostic DNALMs exhibit inconsistent performance and do not offer compelling gains over alternative baseline models for most tasks, despite requiring significantly more computational resources. We discuss potentially promising modeling, data curation, and evaluation strategies for the next generation of DNALMs. Our benchmark datasets and evaluation framework are available at `https://github.com/kundajelab/DART-Eval`

## 1    Introduction

Large Language Models (LLMs) have revolutionized natural language processing [31] by learning complex patterns, contextual relationships, and hierarchical structure within text in a self-supervised manner. In the biological domain, self-supervised language models trained on protein sequences have enabled high-performance tools for protein structure prediction and design [42, 29, 27]. Building on these advances, analogous DNA Language Models (DNALMs) have been recently developed to learn rich representations of genomic DNA sequences from one or more species, aiming to enhance DNA sequence design, functional syntax discovery, and evolutionary analyses [23].

The human genome is 3 billion base pairs of DNA and encodes two main classes of functional elements (See Appendix B for biological background). *Protein coding sequences* of approximately 20,000 genes span around 1.5% of the genome. These sequences encode a dense, information-rich syntax and serve as templates for transcription to RNA and translation to proteins. In contrast, millions of *non-coding regulatory control elements*, estimated to collectively span 5 to 20% of the genome, orchestrate complex programs of gene transcription and translation that define cell state, fate, and response [11]. The DNA sequence of transcriptional regulatory elements typically encode cell-type

38th Conference on Neural Information Processing Systems (NeurIPS 2024) Track on Datasets and Benchmarks.

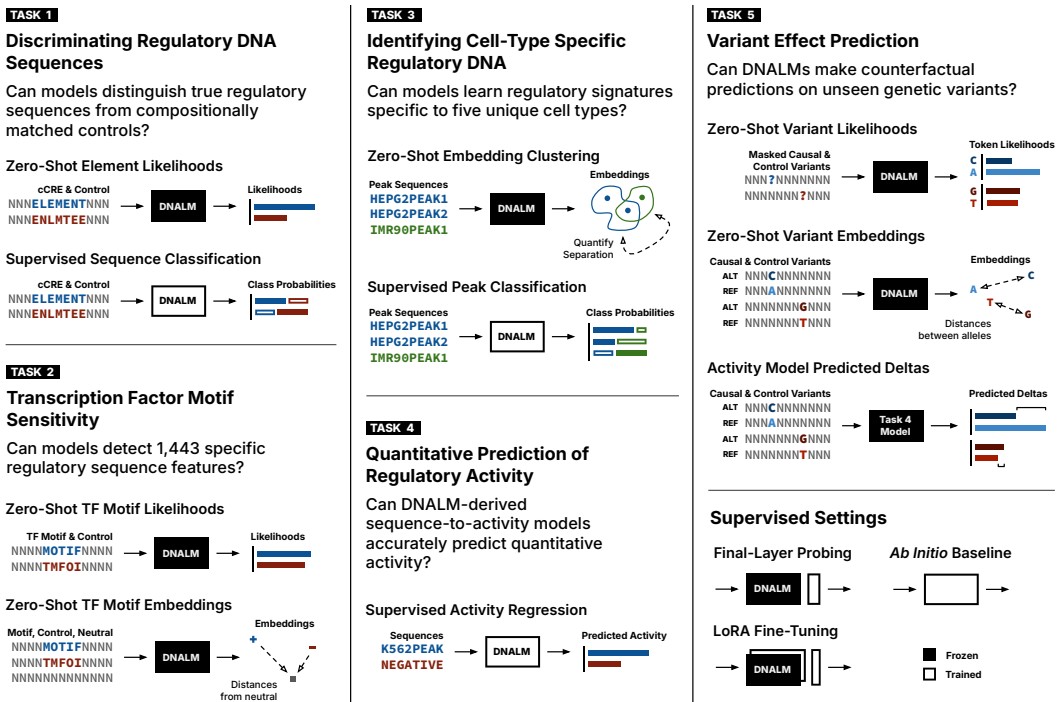

Figure 1: An overview of DART-Eval tasks and settings

specific, sparse, combinatorial syntax involving short, fuzzy DNA motifs bound by regulatory DNA binding proteins (transcription factors, or TFs). The stark contrasts between coding and regulatory sequences, in their syntax, cell-type specificity, and genomic coverage, pose significant challenges for DNALMs to learn optimal representations for diverse downstream applications, necessitating a wide array of benchmarks and evaluation criteria.

Effective DNALMs must, at minimum, achieve three key objectives:

1. Learn representations that can accurately distinguish different types of functional elements.

2. Serve as foundations for training downstream supervised models using potentially sparsely labeled datasets, via probing or fine-tuning, to address salient biological questions and applications.

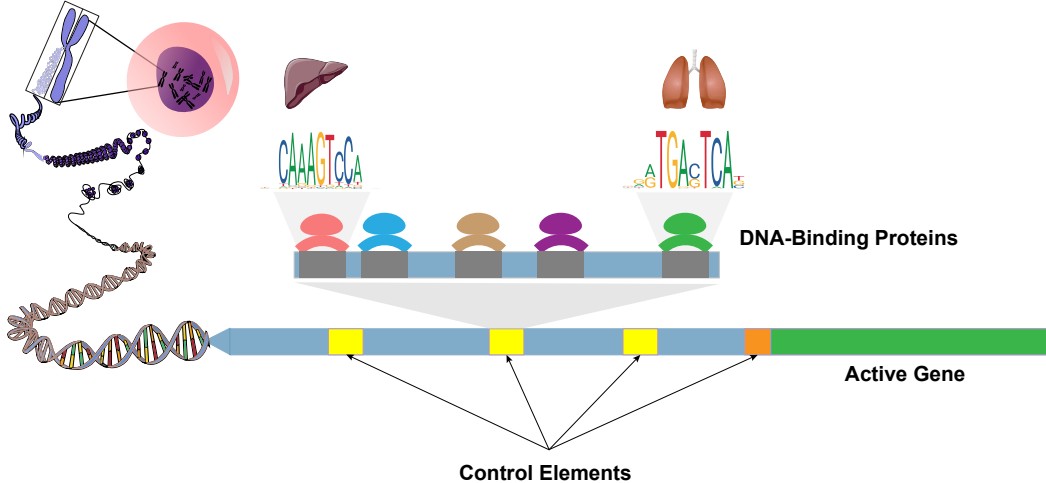

Figure 2: Regulatory DNA syntax is sparse, combinatorial, and cell-type-dependent [4, 3, 2, 38]

3. Outperform alternative *ab initio* models on biologically relevant tasks, either by themselves or as foundations for a supervised model.

The most biologically relevant applications of regulatory sequence models include interpreting and discovering functional regulatory sequences, predicting the effects of sequence perturbations including genetic variants, and designing sequences with desired properties. However, contemporary DNALMs have not been rigorously evaluated against state-of-the-art *ab initio* models for their ability to learn representations of regulatory DNA that enhance performance on these important downstream applications.

To address this gap, we introduce DART-Eval (**DNA R**egula**T**ory **Eval**uations), a suite of benchmarks to assess the utility of regulatory DNA representations learned by DNALMs (Figure 1). We evaluate the performance of contemporary large DNALMs, trained genome-wide without leveraging genomic annotations, across five sets of downstream tasks of increasing difficulty, comparing their zero-shot, probed and fine-tuned performance against state-of-the-art, *ab initio* models.

DART-Eval offers:

- **Five sets of tasks with representative benchmark datasets** that enable evaluation of diverse properties and use cases of regulatory DNA representations learned by DNALMs.
- **An adaptable framework** to prepare embeddings, train probing models, and fine-tune DNALMs.

Our systematic evaluation reveals three key results:

- **Embedding-free methods generally outperform embedding-based methods**.
- **Simpler *ab initio* supervised models match or exceed the performance of much larger, fine-tuned DNALMs.**
- **DNALMs perform particularly poorly on counterfactual prediction tasks and substantially underperform *ab initio* supervised models**.

DART-Eval serves as a resource for developing more effective regulatory DNA models and rigorously assessing the utility of DNALMs.

## 2 Related Work

### 2.1 DNA language models of the human genome

In this work, we specifically evaluate contemporary DNALMs that are trained in an annotation-agnostic manner across the entire human genome and in some cases, additional species. We evaluate Caduceus, DNABERT-2, GENA-LM, HyenaDNA, Mistral-DNA, and Nucleotide Transformer, summarized in Table 1 [39, 47, 16, 34, 5, 13]. We select the most capable version of each model family, with resource requirements summarized in Table S5. These models employ diverse approaches. Pre-training objectives include BERT-style masking and autoregressive next-token-prediction. Tokenization strategies include single-nucleotide, non-overlapping fixed-size *k*-mers, and byte-pair encodings. Model architectures include Transformers (self-attention), Hyena (sub-quadratic-time implicit long convolutions), Mamba (selective state space), and differ in maximum context lengths.

Our current study does not include some other classes of DNALMs, such as those modeling evolutionarily distant organisms [33, 9], annotation-aware models trained on pre-specified classes of genomic regions [23], and models that incorporate additional features like multiple sequence alignments [9]. Our benchmarks can be readily adapted to these models as well, and we aim to include these in future extensions.

### 2.2 Previous DNALM benchmarks

Several DNALM benchmarks have been proposed [34, 13, 16, 47, 30, 41], including evaluations focused on regulatory DNA. However, existing evaluations face three critical limitations. First, they focus on surrogate prediction tasks that do not directly address the main downstream applications (interpretation, counterfactual predictions, design). Second, fundamental flaws in benchmark dataset design undermine evaluations. Lastly, reliance on oversimplified or flawed baseline approaches often exaggerates the relative benefits of DNALMs (summarized in Table 2).

Table 1: An overview of evaluated DNALMs and *ab initio* baselines.

| Model | Variant | Objective | Tokenization | Parameters | Species | Max Context |
|---|---|---|---|---|---|---|
| Caduceus | `ps_131k_d-256_n-16` | Masked | Single nucleotide | 7.7M | Human | 131 kbp |
| DNABERT-2 | `117M` | Masked | Byte-pair | 117M | Multi-Species | 10 kbp* |
| GENA-LM | `bert-large-t2t` | Masked | Byte-pair | 336M | Multi-Species | 4.5 kbp |
| HyenaDNA | `large-1m` | Autoregressive | Single nucleotide | 6.6M | Human | 1 mbp |
| Mistral-DNA | `v1-1.6B-hg38` | Autoregressive | Byte-pair | 1.6B | Human | 10 kbp |
| Nucleotide Transformer[†] | `v2-500m-multi-species` | Masked | $k$-mer | 500M | Multi-species | 12 kbp |
| *Ab initio* Probing-head-like | - | Supervised | Single nucleotide | 68K | Human | 500 bp* |
| *Ab initio* ChromBPNet-like | - | Supervised | Single nucleotide | 5.6M | Human | 500 bp* |
| *Ab initio* ChromBPNet | - | Supervised | Single nucleotide | 6.6M | Human | 2114 bp[‡] |

[*] Maximum evaluated context length. Infinite in theory.

[†] Abbreviated as "NT" for conciseness when necessary.

[‡] Constrained by base-pair-resolution prediction head.

Table 2: Limitations of existing non-coding regulatory DNA evaluations for DNALMs

| | GENA-LM | DNABERT-2 GUE | Nucleotide Transformer | HyenaDNA | BEND | Tang *et al.* | **DART-Eval** |
|---|---|---|---|---|---|---|---|
| Includes distal regulatory elements | ✗ | ✓ | ✓ | ✓ | ✓ | ✓ | ✓ |
| Number of TFs individually analyzed | 3[†] | 10 | - | -[†] | - | 10 | 1443 |
| Regression tasks for quantitative assays | ✗ | ✗ | ✗ | ✗ | ✗ | ✱ | ✓ |
| Compositionally-controlled negatives | ✓ | ✱ | ✱ | ✱ | ✗ | ✗ | ✓ |
| Accounts for LD among variants | - | - | ✗ | - | ✗ | ✓ | ✓ |

For each evaluation, we consider only tasks for non-coding regulation in mammalian species. We find that current evaluations focus on surrogate tasks, often draw biological conclusions using incorrect approaches, and compare with flawed baselines.

(✗), (✓), and (✱) indicate that a criterion is met, not met, and partially met (i.e. for only certain tasks) respectively.

(-) indicates that an evaluation has no relevant tasks for the given criterion.

[†] Models were fine-tuned on ChIP-Seq data from 160 TFs. However, reported statistics on individual TFs were more limited or absent.

Current benchmarks often rely on surrogate predictive tasks that do not directly address key downstream applications. For example, DNALMs are often fine-tuned to classify different classes of regulatory elements like promoters or enhancers and evaluated on the same task [13, 34, 16, 47]. However, the practical value of these models lies in their ability to reveal predictive sequence features or design synthetic sequences for each class, capabilities that most existing benchmarks fail to evaluate.

Dataset design issues, such as the lack of rigorous controls, further compromise evaluations. Regulatory DNA sequences typically have higher G/C nucleotide content than the genomic background. Without compositionally matched background sequences, classifiers may perform well without learning biologically relevant regulatory DNA sequence features [44, 26, 46, 11]. Confounders also affect counterfactual variant effect prediction tasks, where the goal is to use variant effect scores to discriminate functional genetic variants from other background variants. Many evaluations incorrectly use trait-associated variants to define functional variant sets, overlooking that most such associations are correlative rather than causal due to linkage disequilibrium (LD) [43]. Benchmarking datasets must therefore be carefully curated to control for such confounders.

The rapid progress in regulatory genomics in terms of data generation and modeling has rendered many previous cutting-edge resources obsolete. However, many benchmarks often use outdated datasets and baseline models. The latest *ab initio* supervised models are trained on quantitative, high-resolution molecular readouts of regulatory activity across multiple cell types and vastly improve upon previous generation binary classification models [11, 6, 35]. Effective benchmarks must incorporate these state-of-the-art models as baselines.

Recent benchmarks have begun addressing these limitations by incorporating high-quality regulatory profiling experiments across diverse cellular contexts [30] and experimentally validated regulatory genetic variants [41]. Our work complements these efforts with carefully curated benchmark datasets that (1) carefully control for biological confounders, (2) enable evaluation of the model's ability to discover a comprehensive set of regulatory sequence features across diverse cell types, and (3) utilize panels of high-confidence causal variants affecting regulatory activity.

Table 3: Regulatory element identification

| | Zero-Shot | Probed | | Fine-Tuned | | Trained *ab initio* | |
|---|---|---|---|---|---|---|---|
| Model | Accuracy | Absolute Acc. | Paired Acc. | Absolute Acc. | Paired Acc. | Absolute Acc. | Paired Acc. |
| Caduceus | **0.971** | 0.726 | 0.896 | 0.903 | 0.971 | - | - |
| DNABERT-2 | 0.876 | 0.847 | 0.943 | 0.913 | 0.973 | - | - |
| GENA-LM | 0.947 | **0.887** | **0.959** | 0.909 | 0.972 | - | - |
| HyenaDNA | 0.891 | 0.847 | 0.935 | 0.877 | 0.952 | - | - |
| Mistral-DNA | 0.863 | 0.759 | 0.859 | 0.817 | 0.905 | - | - |
| Nucleotide Transformer | 0.745 | 0.819 | 0.917 | **0.920** | **0.976** | - | - |
| *Ab initio* Probing-head-like | - | - | - | - | - | 0.846 | 0.932 |

The best-performing model in each setting is bolded. Here, we evaluate the models' abilities to prioritize curated regulatory elements against matched control sequences. Zero-shot accuracy and supervised paired accuracy quantify the fraction of positives prioritized over their corresponding negatives. Supervised absolute accuracy quantifies the fraction of labels assigned correctly. For this task, (1) DNALMs were effective in a zero-shot setting, (2) final-layer DNALM embeddings offered little additional signal over the raw sequence, and (3) fine-tuned DNALMs offered a moderate improvement over *ab initio* models.

# 3 DNALM evaluation approaches and baselines

This section describes the different types of zero-shot, probed, and fine-tuned DNALM evaluation approaches as well as the baseline models.

**Zero-Shot Analyses** We used two broad strategies to evaluate pre-trained DNALMs in a zero-shot manner without task-specific tuning. First, embedding-based evaluations utilize the mean last-layer embeddings across tokens. Second, likelihood-based evaluations are derived from the cross-entropy loss as the input (interpreted as a negative log-likelihood and closely related to perplexity scores). For autoregressive models, the likelihood is computed from the overall loss of the next-token predictions. For masked models, the quasi-likelihood for a single token is defined as the likelihood of that token, given an input masked at that token. These quasi-likelihoods are then summed across tokens.

**Supervision via probing or fine-tuning** We implemented two efficient approaches for supervised tuning of the pre-trained DNALMs. First, final-layer probing involves training a CNN classifier on the final hidden layer outputs of the DNALM, with the base model frozen. Second, parameter-efficient fine-tuning uses LoRA [20] to train low-rank adapters on all linear and convolutional layers in the base model. The specific architectures used are detailed in Appendix D.2.

*Ab Initio* **Baselines** We compared the DNALMs to supervised baseline models that were trained *ab initio*. First, we used ChromBPNet as the baseline model for regression tasks (Section 4.1) involving chromatin accessibility (a measure of regulatory activity). ChromBPNet is a dilated CNN architecture with a 2 Kb local receptive field that predicts base-resolution signal profiles of chromatin accessibility from regulatory DNA sequence [35]. Second, for the other tasks (Sections 4.2 - 4.5), we used a simple CNN classifier architecturally similar to our probing CNN architecture but with one additional layer. Lastly, for the cell-type-specific sequence classification task (Section 4.3), we used a CNN model similar to ChromBPNet but trained on binary labels. Further details are provided in Appendix D.3.

# 4 Prediction tasks and evaluation results

## 4.1 Distinguishing regulatory DNA from background sequences

First, we designed a relatively easy prediction task that evaluates a model's ability to distinguish high-confidence regulatory element sequences from compositionally matched synthetic control sequences. The positive element set was derived from 2.3 million candidate cis-regulatory elements (cCREs) curated by the ENCODE consortium based on experimental profiling of biochemical regulatory activity [12]. Negative set sequences were generated by shuffling each sequence while maintaining dinucleotide frequencies, thereby destroying syntactic information but preserving key compositional properties [36, 8].

**Data** The dataset for this task consists of 2.3 million cCREs of length 350 bp (Appendix C.1) and an equivalent number of synthetic dinucleotide-shuffled negatives. See Appendix E.1 for preprocessing details.

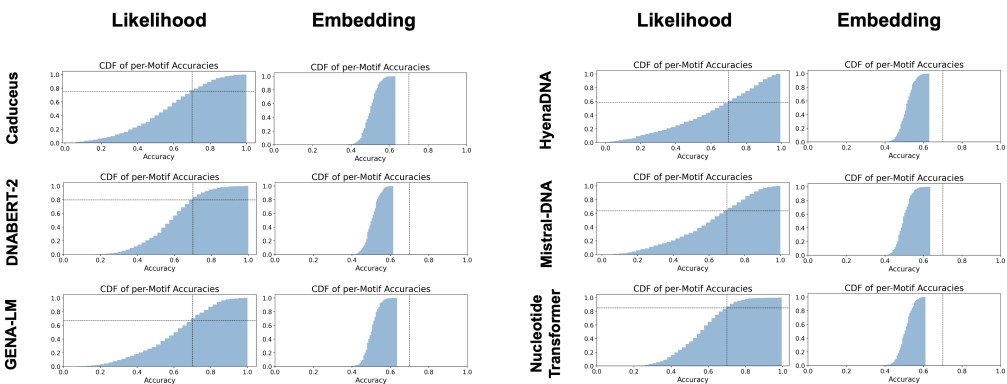

Figure 3: Distributions of zero-shot accuracies across 1443 transcription factor motifs, testing the ability to distinguish motif instances from background sequences. Vertical and horizontal lines represent 70% accuracy thresholds. In the likelihood setting, models identify most but not all motifs. In the embedding settings, models fail to distinguish motifs from background sequences.

**Metrics** In the zero-shot setting, DNALM model likelihoods were obtained for each pair of cCRE and shuffled negative, with a "correct" prediction assigning a greater likelihood to the cCRE than the negative. In supervised settings, the models predict whether a given element is a cCRE or a negative. We define "absolute accuracy" as the fraction of elements and controls classified correctly, and "paired accuracy" as the fraction of pairs where the model assigns a greater probability to cCREs in the positive set than the synthetic controls in the negative set.

**Results** All DNALMs prioritized cCREs over synthetic negative control sequences in a zero-shot setting (Table 3), suggesting that the models are likely learning at least some regulatory sequence features beyond compositional biases. Probing models demonstrated similar absolute accuracies and improved paired accuracies, compared to the zero-shot setting. Fine-tuning yielded a further improvement in prioritization, with paired accuracies approaching 1. The *ab initio* CNN baseline model, in comparison, demonstrated similar performance to the probed models.

## 4.2 Assessing sensitivity to known regulatory sequence motifs

Next, we evaluated whether the DNALMs had learned sequence features that are known to drive regulatory activity. Transcriptional regulatory elements encode one or more short sequence motifs that recruit sequence-specific regulatory DNA binding proteins called transcription factors (TFs). Here, we assess the models' abilities to distinguish known TF binding motifs from matched shuffled negative control motifs.

**Data** We individually tested known consensus binding motifs of 1443 TFs from the HOCOMOCO v12 database [45], further described in Appendix C.2. We derived 100 neutral backgrounds from dinucleotide-shuffled ENCODE cCRE sequences. Positive sequences were constructed by inserting TF motifs at the center of each background sequence. Negative sequences were similarly constructed by inserting shuffled motifs. Both forward and reverse complements of each positive and negative sequence were scored through the models and used for evaluation.

**Metrics** We evaluated the DNALMs in a zero-shot setting by calculating likelihoods and embeddings. For the likelihood-based approach, we considered a prediction to be "correct" if the model assigned a higher likelihood to a positive sequence than its corresponding negative sequence. For the embedding-based approach, we defined a correct prediction as one where the cosine embedding distance from the neutral background sequence to its corresponding positive sequence was greater than the distance from the neutral sequence to a corresponding negative sequence. Accuracies were computed individually for each TF motif (Appendix E.2).

**Results** In the likelihood setting, all DNALMs were able to prioritize positive sequences containing TF motifs over negative controls (Figure 3). However, accuracies varied substantially across motifs, suggesting that some motifs are likely not encoded in the internal representations learned by the DNALMs (Table S7). Notably, motif likelihood scores were highly correlated between models,

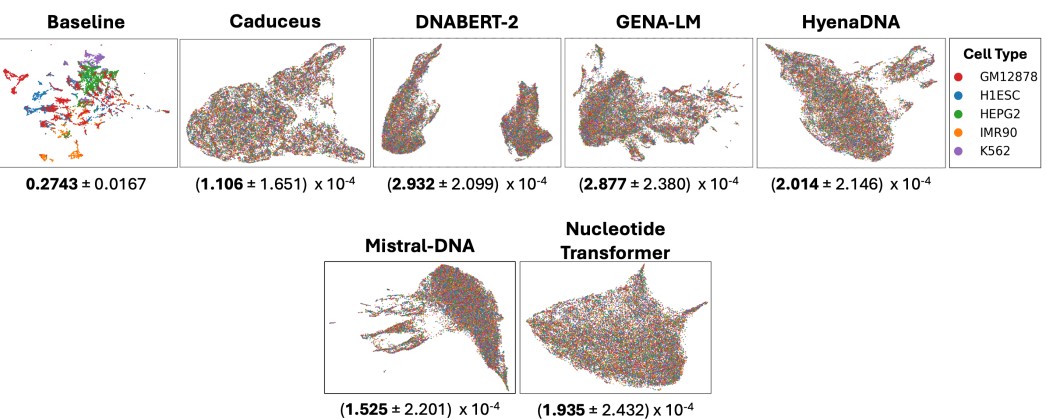

Figure 4: UMAP of model embeddings for sequences with experimentally identified cell-type-specific activity, colored by the true labels. The baseline embedding is a vector of canonical motif instances identified by FIMO. Numerical values are adjusted mutual information scores between true labels and a $k$-means clustering with $k = 50$, along with a 95% confidence interval across clustering seeds, measuring the clustering quality w.r.t. the true labels. Only the baseline model yielded a useful embedding space for distinguishing sequence features for cell-type-specific activity.

suggesting a strong influence of the underlying pre-training data distribution on performance (Figures S2, S3). These trends were also preserved when motifs were aggregated over TFs belonging to the same families based on the similarity of their DNA binding domains and motifs (Figures S4, S5). In contrast, no model reliably prioritized motifs in the embedding setting, with the vast majority of accuracies falling between 40% and 60% (Figure 3). These results indicate that final-layer averaged embeddings and cosine distance measurements do not fully capture the models' expressivity. Lastly, motif discovery performance (this task) was nearly always lower than performance for discriminating regulatory elements (Section 4.1). We hypothesize that this discrepancy stems from two key factors. First, DNALMs learn only a subset of TF motifs, primarily capturing those that appear frequently across the genome. Second, regulatory elements contain multiple TF binding sites, allowing successful discrimination even with an incomplete repertoire of learned motifs.

## 4.3 Learning cell-type-specific regulatory sequence features

Cell-type identity emerges from distinct patterns of regulatory element activity across a shared genome. Hence, we evaluated whether representations learned by DNALMs encode cell-type specific regulatory sequence features.

**Data** We curated cell-type specific regulatory elements across five diverse cell lines based on ATAC-seq chromatin accessibility experiments that highlight regulatory regions bound by TFs ([10]). Specifically, we used regions with strong ATAC-seq accessibility signal (peaks) as candidate regulatory elements in each cell line C.3. We then used DESeq2 for differential analysis to identify cell-type specific elements that showed significantly higher chromatin accessibility in exactly one cell type relative to the others [28] (Appendix E.3).

**Metrics** We evaluated models in zero-shot and supervised settings. For zero-shot evaluation, we clustered model embeddings of the cell-type specific regulatory sequences using $k$-means and quantified label separation using the adjusted Mutual Information Score across labels [1]. In the supervised setting, we evaluate the performance of classification of the cell-type specific regulatory sequences using overall accuracy and binary classification metrics (accuracy, AUROC, AUPRC) for each cell type versus the others (Appendix E.3).

**Results** In the zero-shot setting, all DNALMs showed poor cell-type separation in their embeddings when compared to a simple baseline approach that used $k$-means on the sequences represented using motif counts of known motifs (Figure 4) (Appendix E.3). In the supervised setting, the *ab initio* CNN baseline generally matched or outperformed all other models. Fine-tuned models outperform probed models, which performed comparably to a smaller *ab initio* CNN baseline (Tables 4, S8 - S13).

Table 4: Cell-type-specific element classification.

| | | Overall | GM12878 | H1ESC | HEPG2 | IMR90 | K562 |
|---|---|---|---|---|---|---|---|
| Setting | Model | Accuracy | AUROC | AUROC | AUROC | AUROC | AUROC |
| Probed | Caduceus | 0.281 | 0.535 | 0.622 | 0.680 | 0.576 | 0.587 |
| | DNABERT-2 | 0.371 | 0.652 | 0.757 | 0.762 | 0.691 | 0.691 |
| | GENA-LM | 0.383 | 0.627 | 0.787 | 0.773 | 0.714 | 0.693 |
| | HyenaDNA | 0.587 | 0.849 | 0.889 | 0.862 | 0.882 | 0.799 |
| | Mistral-DNA | 0.329 | 0.582 | 0.678 | 0.723 | 0.643 | 0.646 |
| | Nucleotide Transformer | 0.420 | 0.744 | 0.795 | 0.783 | 0.779 | 0.711 |
| Fine-Tuned | Caduceus | **0.671** | 0.900 | **0.937** | **0.901** | **0.929** | **0.878** |
| | DNABERT-2 | 0.650 | 0.894 | 0.930 | 0.891 | 0.922 | 0.871 |
| | GENA-LM | 0.636 | 0.877 | 0.923 | 0.887 | 0.911 | 0.862 |
| | HyenaDNA | 0.610 | 0.875 | 0.906 | 0.873 | 0.908 | 0.847 |
| | Mistral-DNA | 0.402 | 0.687 | 0.762 | 0.734 | 0.748 | 0.710 |
| | Nucleotide Transformer | 0.632 | 0.880 | 0.925 | 0.881 | 0.920 | 0.867 |
| Ab initio | Probing-head-like | 0.474 | 0.754 | 0.836 | 0.757 | 0.807 | 0.741 |
| | ChromBPNet-like | 0.667 | **0.903** | 0.929 | 0.894 | 0.921 | 0.848 |

We underline the best-performing model for each setting and bold the best-performing model across all settings. For each model, we evaluated overall accuracy across all classes and AUROC between each class and the remainder. We see that *ab initio* sequence models performed comparably to the best fine-tuned DNALMs and substantially outperformed all probed DNALMs.

## 4.4 Predicting quantitative measures of regulatory activity from sequence

ATAC-seq and DNase-seq experiments performed in a cell type of interest provide genome-wide quantitative measurements of chromatin accessibility [10, 40]. The quantitative chromatin accessibility signal of a regulatory element is dictated by the repertoire of TFs that bind specific syntax of sequence motifs encoded in its sequence. We therefore evaluated whether representations learned by DNALMs enable accurate prediction of quantitative chromatin accessibility.

**Data** We trained sequence-to-activity (S2A) regression models to predict quantitative DNase-seq signal (measured as log(counts of sequencing reads)) over 2 Kb genomic sequences. The input sequences included DNase-seq peaks (regions with a strong, statistically significant signal) and other G/C-content-matched background genomic regions. Models were trained and evaluated separately on DNase-seq data from five cell types as in Section 4.3 (Appendix C.3).

**Metrics** We assessed regression performance using Pearson and Spearman correlation between predicted and observed signal, evaluated across peak regions only and across a union of peak and background regions. Additionally, we computed binary classification metrics (AUROC and AUPRC), using a positive set of high-confidence, reproducible peaks and a negative set of background regions (Appendix E.4).

**Results** Fine-tuned DNALM S2A models showed strong performance, with the strongest models matching or moderately outperforming the *ab-initio* baseline CNN (ChromBPNet) model on regression and classification metrics (Tables 5, S14 - S18). In contrast, probed models substantially underperformed ChromBPNet, indicating that frozen last-layer embeddings lack sufficient expressivity for accurate prediction of quantitative accessibility.

## 4.5 Predicting counterfactual effects of regulatory genetic variants

A critical challenge in human genetics is predicting how genetic variants affect gene regulation through changes in chromatin accessibility. Models trained to predict regulatory activity from sequence (S2A models) (such as those in Section 4.4) are typically used in a counterfactual setting to predict the effects of genetic variants on regulatory activity. This is a particularly challenging task since the S2A models are never directly trained on genetic variation data. We evaluated the ability of DNALMs to prioritize and predict the quantitative effects of regulatory genetic variants that impact chromatin accessibility.

**Data** We utilized data from two quantitative trait locus (QTL) mapping studies that associate genetic variation with variation of chromatin accessibility from ATAC-seq or DNase-seq experiments across a large cohort of lymphoblastoid cell lines (LCLs) from individuals of African ancestry [15, 14] (Appendix C.4). These datasets identify genetic variants with statistically significant effects on chromatin accessibility as caQTLs (for ATAC-seq) and dsQTLs (for DNase-seq). We enrich for likely

Table 5: Chromatin activity prediction

| Setting | Model | Spearman $r$ among positives | | | | | AUROC (positives vs. negatives) | | | | |
|---|---|---|---|---|---|---|---|---|---|---|---|
| | | GM12878 | H1ESC | HEPG2 | IMR90 | K562 | GM12878 | H1ESC | HEPG2 | IMR90 | K562 |
| Probed | Caduceus | 0.251 | 0.371 | 0.312 | 0.149 | 0.401 | 0.605 | 0.608 | 0.611 | 0.610 | 0.616 |
| | DNABERT-2 | 0.395 | 0.584 | 0.357 | 0.275 | 0.483 | 0.757 | 0.763 | 0.650 | 0.729 | 0.721 |
| | GENA-LM | 0.490 | 0.678 | 0.401 | 0.329 | 0.461 | 0.784 | 0.809 | 0.771 | 0.799 | 0.761 |
| | HyenaDNA | 0.362 | 0.538 | 0.345 | 0.237 | 0.438 | 0.708 | 0.728 | 0.641 | 0.702 | 0.662 |
| | Mistral-DNA | 0.293 | 0.500 | 0.349 | 0.244 | 0.431 | 0.586 | 0.644 | 0.653 | 0.712 | 0.678 |
| | NT | 0.410 | 0.595 | 0.337 | 0.270 | 0.499 | 0.757 | 0.765 | 0.648 | 0.739 | 0.764 |
| Fine-Tuned | Caduceus | 0.503 | 0.744 | 0.454 | 0.479 | 0.570 | 0.935 | 0.954 | 0.896 | **0.976** | 0.933 |
| | DNABERT-2 | 0.489 | 0.717 | 0.472 | 0.470 | 0.529 | 0.916 | 0.940 | 0.893 | 0.963 | 0.917 |
| | GENA-LM | 0.467 | 0.696 | 0.439 | 0.421 | 0.532 | 0.908 | 0.942 | 0.878 | 0.961 | 0.910 |
| | HyenaDNA | 0.435 | 0.672 | 0.406 | 0.426 | 0.446 | 0.853 | 0.927 | 0.854 | 0.941 | 0.750 |
| | Mistral-DNA | 0.372 | 0.573 | 0.360 | 0.302 | 0.430 | 0.789 | 0.838 | 0.731 | 0.855 | 0.796 |
| | NT | 0.515 | 0.737 | 0.513 | 0.489 | **0.583** | 0.938 | **0.958** | **0.922** | 0.975 | **0.941** |
| *Ab initio* | ChromBPNet | **0.540** | **0.754** | **0.534** | **0.549** | 0.574 | **0.940** | 0.952 | 0.910 | 0.975 | 0.917 |

We underline the best-performing model for each setting and bold the best-performing model across all settings. For each model, we evaluated the correlation between predicted and true signals across peak regions. Additionally, we evaluated classification performance against a positive set of high-confidence peak regions and a negative set of background sequences. We see that DNALM-derived models do not offer a consistent advantage over the *ab initio* baseline models.

causal caQTLs and dsQTLs by restricting to those that fall within accessible regulatory elements (peak regions). These variants form the positive set. The negative set consists of other background genetic variants in peak regions that do not exhibit statistically significant associations with chromatin accessibility. Each genetic variant consists of a pair of alleles (two different nucleotides) $\in$ {A, C, G, T} and a label $y \in$ {0,1} with 0 = background and 1 = significant.

**Metrics** We extracted the 2 Kb genomic sequence context of each variant and scored two versions of the sequence that contain each of the alleles of the variant with different models. We scored variants in both zero-shot and supervised settings. For embedding-based zero-shot scoring, we computed the cosine distance between the two variant allele sequences. For likelihood-based zero-shot scoring, we computed the log-likelihood difference using an input sequence with the variant-containing token masked and taking the loss with respect to the two alleles. The likelihood-based evaluations were only conducted for models with fixed encodings (Nucleotide Transformer and HyenaDNA) since changing a single base can affect the entirety of a byte-pair encoding. For supervised scoring, we used the chromatin accessibility S2A models from Section 4.4 trained on a single LCL sample (GM12878). We computed the absolute difference in predicted accessibility between the two alleles. For each dataset and variant effect score, we computed the AUROC and AUPRC with respect to the positive and negative variant sets. In addition, we computed the Pearson correlation between the reported effect size of association from the QTL study and the predicted activity difference for the supervised models (Appendix E.5).

**Results** In the zero-shot setting, Nucleotide Transformer achieved the best performance for both embedding and likelihood-based approaches (Table 6). In supervised evaluation, fine-tuned sequence-to-activity models substantially outperformed their probed counterparts. However, despite matching ChromBPNet's performance in chromatin accessibility prediction (Section 4.4), fine-tuned models underperformed the *ab-initio* baseline ChromBPNet in variant effect prediction. This discrepancy highlights the critical importance of including counterfactual tasks in evaluations alongside observational assessments (Tables 6, S19, and S20).

## 5   Discussion

In this study, we present DART-Eval, a suite of representative benchmark datasets for evaluating regulatory DNA representations learned by DNALMs. Our evaluations spans five tasks, comparing state-of-the-art DNALMs in zero-shot, probed, and fine-tuned settings against strong *ab initio* baseline models. The tasks increase in difficulty from detecting regulatory sequences, to regulatory motif discovery, quantitative prediction of regulatory activity, and finally counterfactual prediction of regulatory genetic variants. While DNALMs excel at simpler tasks, their performance deteriorates with increasing task complexity, highlighting the need for rigorous evaluations to accurately assess the capabilities of these models.

Table 6: Variant scoring

| Dataset | Model | Zero-Shot AUROC | | Probed | | Fine-tuned | | *Ab initio* | |
|---|---|---|---|---|---|---|---|---|---|
| | | Likelihood | Embedding | Pearson $r$ | AUROC | Pearson $r$ | AUROC | Pearson $r$ | AUROC |
| African | Caduceus | 0.525 | 0.519 | -0.004 | 0.512 | 0.259 | 0.650 | | |
| | DNABERT-2 | - | 0.480 | 0.007 | 0.502 | 0.184 | 0.616 | - | - |
| | GENA-LM | - | 0.508 | -0.007 | 0.515 | 0.201 | 0.604 | - | - |
| | HyenaDNA | 0.486 | 0.515 | 0.012 | 0.566 | 0.265 | 0.611 | - | - |
| | Mistral-DNA | - | 0.520 | 0.018 | 0.502 | 0.085 | 0.510 | | |
| | NT | 0.525 | 0.519 | 0.006 | 0.525 | 0.230 | 0.623 | - | - |
| | ChromBPNet | - | - | - | - | - | - | **0.671** | **0.772** |
| Yoruban | Caduceus | 0.443 | 0.508 | 0.017 | 0.490 | 0.513 | 0.666 | | |
| | DNABERT-2 | - | 0.505 | 0.024 | 0.476 | 0.473 | 0.631 | - | - |
| | GENA-LM | - | 0.501 | 0.059 | 0.466 | 0.414 | 0.628 | - | - |
| | HyenaDNA | 0.436 | 0.515 | -0.042 | 0.467 | 0.503 | 0.573 | - | - |
| | Mistral-DNA | - | 0.475 | -0.003 | 0.432 | 0.053 | 0.504 | | |
| | NT | 0.469 | 0.613 | 0.129 | 0.516 | 0.507 | 0.670 | - | - |
| | ChromBPNet | - | - | - | - | - | - | **0.738** | **0.892** |

We underline the best-performing model for each setting and bold the best-performing model across all settings. In zero-shot settings, allelic effects of variants were scored by measuring the difference in model-derived embeddings or likelihoods of for sequences containing each allele of the variant. We computed classification metrics between positive and control variant sets, expecting positive variants to have larger predicted allelic effects. Here, unlike the other zero-shot tasks, the likelihood setting did not substantially outperform the embedding setting. In supervised settings, for the positive variants, we computed the correlation between measured and predicted allelic effects. We also computed classification metrics (AUROC and AUPRC) relative to the positive and negativevariant sets. The *ab-initio* baseline CNN (ChromBPNet) substantially outperformed DNALM-based methods.

Although DNALMs successfully discriminate regulatory DNA from background sequences, they appear to learn incomplete repertoires of regulatory sequence features. This limitation likely stems from the sparsity and the uneven distribution of regulatory features; regulatory elements constitute only 10-20% of the human genome, and certain classes of regulatory features occur at substantially different frequencies. Potential strategies to address this challenge include balanced sampling of training examples across different classes of functional elements, incorporating regulatory annotations as tokens, or training on subsets of the genome that are functionally related (e.g., sets of candidate regulatory elements) rather than across the entire genome.

Our analysis reveals several critical insights about DNALM architecture and modeling choices. Consistent with previous studies, we observe that embedding-derived approaches (e.g. embedding distance, final-layer probing) generally underperform methods that leverage models' full expressivity (e.g. likelihoods, fine-tuning) [32, 13]. For likelihood-based methods, masked objectives appear to be less efficient than autoregressive objectives due to required iterative token masking. Byte-pair encodings pose a particular challenge for variant effect prediction, as single-base changes can alter multiple tokens in unpredictable ways, making it difficult to compare sequence likelihoods. While fine-tuning generally achieves superior performance relative to probing, it demands significantly more computational resources, requiring gradient backpropagation through the entire model.

While DART-Eval offers a rigorous framework for the evaluation of regulatory DNA representations learned by DNALMs, future extensions could enhance its scope. Our current evaluations are limited to tasks involving short, local sequence contexts, and do not encompass tasks that require long-range context, such as the prediction of distal regulatory interactions, gene expression, or 3D genome architecture. With continued improvement in functional annotation of genomes of other model organisms, benchmarking DNALMs on diverse species will become increasingly relevant for assessing the generalizability of learned representations. Expanding task diversity to cover a more comprehensive range of regulatory elements (e.g. 5'-UTRs, 3'-UTRs, and splice sites), and incorporating evaluations related to transcriptional and post-transcriptional regulatory mechanisms would enable a more complete assessment of regulatory function coverage.

Overall, DART-Eval establishes a foundational benchmark for assessing DNALMs, with the potential to drive future advances in the development of DNALMs and their applications for regulatory genomics.

## Acknowledgments and Disclosure of Funding

The authors acknowledge funding support from NIH grants 5U24HG007234, U01HG009431, and U01HG012069 to AK. AS is supported in part by the National Science Foundation Graduate Research Fellowship (NSF GRFP). AW and A Pampari are supported by the Stanford BioX Fellowship. We thank Alex Tseng for providing a dinucleotide shuffling algorithm, Salil Deshpande for providing an algorithm for constructing consensus peak sets across multiple experiments, and Jacob Schreiber for providing a reference PyTorch implementation of ChromBPNet.

## References

[1] adjusted_mutual_info_score —scikit-learn 1.5.0 documentation.

[2] Dna and chromosomes (available under https://pixabay.com/service/license-summary/). `https://pixabay.com/vectors/genetics-chromosomes-rna-dna-156404/`.

[3] Liver (available under https://creativecommons.org/licenses/by-sa/3.0/deed.en). `https://commons.wikimedia.org/wiki/File:Digestive_system_-_Liver_1_--_Smart-Servier.png`.

[4] Lung (available under https://creativecommons.org/licenses/by/4.0/deed.en). `https://commons.wikimedia.org/wiki/File:201405_lung.png`.

[5] Mistral-DNA: Mistral model for genomics |by raphael mourad |medium.

[6] Z. Avsec, V. Agarwal, D. Visentin, J. R. Ledsam, A. Grabska-Barwinska, K. R. Taylor, Y. Assael, J. Jumper, P. Kohli, and D. R. Kelley. Effective gene expression prediction from sequence by integrating long-range interactions. *Nature Methods*, 18(10):1196–1203, oct 2021.

[7] Z. Avsec, M. Weilert, A. Shrikumar, S. Krueger, A. Alexandari, K. Dalal, R. Fropf, C. McAnany, J. Gagneur, A. Kundaje, and J. Zeitlinger. Base-resolution models of transcription-factor binding reveal soft motif syntax. *Nature Genetics*, 53(3):354–366, mar 2021.

[8] T. L. Bailey, J. Johnson, C. E. Grant, and W. S. Noble. The MEME suite. *Nucleic Acids Research*, 43(W1):W39–49, jul 2015.

[9] G. Benegas, C. Albors, A. J. Aw, C. Ye, and Y. S. Song. GPN-MSA: an alignment-based DNA language model for genome-wide variant effect prediction. *BioRxiv*, apr 2024.

[10] J. D. Buenrostro, B. Wu, H. Y. Chang, and W. J. Greenleaf. ATAC-seq: a method for assaying chromatin accessibility genome-wide. *Current Protocols in Molecular Biology*, 109:21.29.1–21.29.9, jan 2015.

[11] E. P. Consortium. An integrated encyclopedia of DNA elements in the human genome. *Nature*, 489(7414):57–74, sep 2012.

[12] E. P. Consortium, J. E. Moore, M. J. Purcaro, H. E. Pratt, C. B. Epstein, N. Shoresh, J. Adrian, T. Kawli, C. A. Davis, A. Dobin, R. Kaul, J. Halow, E. L. Van Nostrand, P. Freese, D. U. Gorkin, Y. Shen, Y. He, M. Mackiewicz, F. Pauli-Behn, B. A. Williams, A. Mortazavi, C. A. Keller, X.-O. Zhang, S. I. Elhajjajy, J. Huey, D. E. Dickel, V. Snetkova, X. Wei, X. Wang, J. C. Rivera-Mulia, J. Rozowsky, J. Zhang, S. B. Chhetri, J. Zhang, A. Victorsen, K. P. White, A. Visel, G. W. Yeo, C. B. Burge, E. Lécuyer, D. M. Gilbert, J. Dekker, J. Rinn, E. M. Mendenhall, J. R. Ecker, M. Kellis, R. J. Klein, W. S. Noble, A. Kundaje, R. Guigó, P. J. Farnham, J. M. Cherry, R. M. Myers, B. Ren, B. R. Graveley, M. B. Gerstein, L. A. Pennacchio, M. P. Snyder, B. E. Bernstein, B. Wold, R. C. Hardison, T. R. Gingeras, J. A. Stamatoyannopoulos, and Z. Weng. Expanded encyclopaedias of DNA elements in the human and mouse genomes. *Nature*, 583(7818):699–710, jul 2020.

[13] H. Dalla-Torre, L. Gonzalez, J. Mendoza Revilla, N. Lopez Carranza, A. Henryk Grywaczewski, F. Oteri, C. Dallago, E. Trop, H. Sirelkhatim, G. Richard, M. Skwark, K. Beguir, M. Lopez, and T. Pierrot. The nucleotide transformer: building and evaluating robust foundation models for human genomics. *BioRxiv*, jan 2023.

[14] J. F. Degner, A. A. Pai, R. Pique-Regi, J.-B. Veyrieras, D. J. Gaffney, J. K. Pickrell, S. De Leon, K. Michelini, N. Lewellen, G. E. Crawford, M. Stephens, Y. Gilad, and J. K. Pritchard. DNase i sensitivity QTLs are a major determinant of human expression variation. *Nature*, 482(7385):390–394, feb 2012.

[15] M. K. DeGorter, P. C. Goddard, E. Karakoc, S. Kundu, S. M. Yan, D. Nachun, N. Abell, M. Aguirre, T. Carstensen, Z. Chen, M. Durrant, V. R. Dwaracherla, K. Feng, M. J. Gloudemans, N. Hunter, M. P. S. Moorthy, C. Pomilla, K. B. Rodrigues, C. J. Smith, K. S. Smith, R. A. Ungar, B. Balliu, J. Fellay, P. Flicek, P. J. McLaren, B. Henn, R. C. McCoy, L. Sugden, A. Kundaje, M. S. Sandhu, D. Gurdasani, and S. B. Montgomery. Transcriptomics and chromatin accessibility in multiple african population samples. *BioRxiv*, nov 2023.

[16] V. Fishman, Y. Kuratov, M. Petrov, A. Shmelev, D. Shepelin, N. Chekanov, O. Kardymon, and M. Burtsev. GENA-LM: A family of open-source foundational models for long DNA sequences. *BioRxiv*, jun 2023.

[17] O. Fornes, J. A. Castro-Mondragon, A. Khan, R. van der Lee, X. Zhang, P. A. Richmond, B. P. Modi, S. Correard, M. Gheorghe, D. Baranašić, W. Santana-Garcia, G. Tan, J. Chèneby, B. Ballester, F. Parcy, A. Sandelin, B. Lenhard, W. W. Wasserman, and A. Mathelier. JASPAR 2020: update of the open-access database of transcription factor binding profiles. *Nucleic Acids Research*, 48(D1):D87–D92, jan 2020.

[18] C. E. Grant, T. L. Bailey, and W. S. Noble. FIMO: scanning for occurrences of a given motif. *Bioinformatics*, 27(7):1017–1018, apr 2011.

[19] S. Heinz, C. Benner, N. Spann, E. Bertolino, Y. C. Lin, P. Laslo, J. X. Cheng, C. Murre, H. Singh, and C. K. Glass. Simple combinations of lineage-determining transcription factors prime cis-regulatory elements required for macrophage and b cell identities. *Molecular Cell*, 38(4):576–589, may 2010.

[20] E. J. Hu, Y. Shen, P. Wallis, Z. Allen-Zhu, Y. Li, S. Wang, L. Wang, and W. Chen. LoRA: Low-rank adaptation of large language models. *arXiv*, 2021.

[21] K. Jaganathan, S. Kyriazopoulou Panagiotopoulou, J. F. McRae, S. F. Darbandi, D. Knowles, Y. I. Li, J. A. Kosmicki, J. Arbelaez, W. Cui, G. B. Schwartz, E. D. Chow, E. Kanterakis, H. Gao, A. Kia, S. Batzoglou, S. J. Sanders, and K. K.-H. Farh. Predicting splicing from primary sequence with deep learning. *Cell*, 176(3):535–548.e24, jan 2019.

[22] J. Jumper, R. Evans, A. Pritzel, T. Green, M. Figurnov, O. Ronneberger, K. Tunyasuvunakool, R. Bates, A. Žídek, A. Potapenko, A. Bridgland, C. Meyer, S. A. A. Kohl, A. J. Ballard, A. Cowie, B. Romera-Paredes, S. Nikolov, R. Jain, J. Adler, T. Back, S. Petersen, D. Reiman, E. Clancy, M. Zielinski, M. Steinegger, M. Pacholska, T. Berghammer, S. Bodenstein, D. Silver, O. Vinyals, A. W. Senior, K. Kavukcuoglu, P. Kohli, and D. Hassabis. Highly accurate protein structure prediction with AlphaFold. *Nature*, 596(7873):583–589, aug 2021.

[23] A. Karollus, J. Hingerl, D. Gankin, M. Grosshauser, K. Klemon, and J. Gagneur. Species-aware DNA language models capture regulatory elements and their evolution. *Genome Biology*, 25(1):83, apr 2024.

[24] J. K. Leman, B. D. Weitzner, S. M. Lewis, J. Adolf-Bryfogle, N. Alam, R. F. Alford, M. Aprahamian, D. Baker, K. A. Barlow, P. Barth, B. Basanta, B. J. Bender, K. Blacklock, J. Bonet, S. E. Boyken, P. Bradley, C. Bystroff, P. Conway, S. Cooper, B. E. Correia, B. Coventry, R. Das, R. M. De Jong, F. DiMaio, L. Dsilva, R. Dunbrack, A. S. Ford, B. Frenz, D. Y. Fu, C. Geniesse, L. Goldschmidt, R. Gowthaman, J. J. Gray, D. Gront, S. Guffy, S. Horowitz, P.-S. Huang, T. Huber, T. M. Jacobs, J. R. Jeliazkov, D. K. Johnson, K. Kappel, J. Karanicolas, H. Khakzad, K. R. Khar, S. D. Khare, F. Khatib, A. Khramushin, I. C. King, R. Kleffner, B. Koepnick, T. Kortemme, G. Kuenze, B. Kuhlman, D. Kuroda, J. W. Labonte, J. K. Lai, G. Lapidoth, A. Leaver-Fay, S. Lindert, T. Linsky, N. London, J. H. Lubin, S. Lyskov, J. Maguire, L. Malmström, E. Marcos, O. Marcu, N. A. Marze, J. Meiler, R. Moretti, V. K. Mulligan, S. Nerli, C. Norn, S. Ó'Conchúir, N. Ollikainen, S. Ovchinnikov, M. S. Pacella, X. Pan, H. Park, R. E. Pavlovicz, M. Pethe, B. G. Pierce, K. B. Pilla, B. Raveh, P. D. Renfrew, S. S. R. Burman, A. Rubenstein, M. F. Sauer, A. Scheck, W. Schief, O. Schueler-Furman, Y. Sedan, A. M. Sevy, N. G. Sgourakis, L. Shi, J. B.

Siegel, D.-A. Silva, S. Smith, Y. Song, A. Stein, M. Szegedy, F. D. Teets, S. B. Thyme, R. Y.-R. Wang, A. Watkins, L. Zimmerman, and R. Bonneau. Macromolecular modeling and design in rosetta: recent methods and frameworks. *Nature Methods*, 17(7):665–680, jul 2020.

[25] Q. Li, J. B. Brown, H. Huang, and P. J. Bickel. Measuring reproducibility of high-throughput experiments. *The annals of applied statistics*, 5(3):1752–1779, sep 2011.

[26] R. Li and J. Ernst. Identifying associations of de novo noncoding variants with autism through integration of gene expression, sequence and sex information. *BioRxiv*, mar 2024.

[27] Z. Lin, H. Akin, R. Rao, B. Hie, Z. Zhu, W. Lu, N. Smetanin, R. Verkuil, O. Kabeli, Y. Shmueli, A. Dos Santos Costa, M. Fazel-Zarandi, T. Sercu, S. Candido, and A. Rives. Evolutionary-scale prediction of atomic-level protein structure with a language model. *Science*, 379(6637):1123–1130, mar 2023.

[28] M. I. Love, W. Huber, and S. Anders. Moderated estimation of fold change and dispersion for RNA-seq data with DESeq2. *Genome Biology*, 15(12):550, 2014.

[29] A. Madani, B. Krause, E. R. Greene, S. Subramanian, B. P. Mohr, J. M. Holton, J. L. Olmos, C. Xiong, Z. Z. Sun, R. Socher, J. S. Fraser, and N. Naik. Large language models generate functional protein sequences across diverse families. *Nature Biotechnology*, 41(8):1099–1106, aug 2023.

[30] F. I. Marin, F. Teufel, M. Horrender, D. Madsen, D. Pultz, O. Winther, and W. Boomsma. BEND: Benchmarking DNA language models on biologically meaningful tasks. *arXiv*, 2023.

[31] S. Minaee, T. Mikolov, N. Nikzad, M. Chenaghlu, R. Socher, X. Amatriain, and J. Gao. Large language models: A survey. *arXiv*, 2024.

[32] N. NaderiAlizadeh and R. Singh. Aggregating residue-level protein language model embeddings with optimal transport. *BioRxiv*, jan 2024.

[33] E. Nguyen, M. Poli, M. G. Durrant, A. W. Thomas, B. Kang, J. Sullivan, M. Y. Ng, A. Lewis, A. Patel, A. Lou, S. Ermon, S. A. Baccus, T. Hernandez-Boussard, C. Re, P. D. Hsu, and B. L. Hie. Sequence modeling and design from molecular to genome scale with evo. *bioRxiv*, jan 2024.

[34] E. Nguyen, M. Poli, M. Faizi, A. Thomas, C. Birch-Sykes, M. Wornow, A. Patel, C. Rabideau, S. Massaroli, Y. Bengio, S. Ermon, S. A. Baccus, and C. Ré. HyenaDNA: Long-range genomic sequence modeling at single nucleotide resolution. *arXiv*, nov 2023.

[35] A. Pampari, A. Shcherbina, S. Nair, A. Wang, A. Patel, K. Mualim, S. Kundu, and A. Kundaje. Bias factorized, base-resolution deep learning models of chromatin accessibility reveal cis-regulatory sequence syntax, transcription factor footprints and regulatory variants. *Biorxiv (*https://zenodo.org/record/7567628), 2024.

[36] E. Prakash, A. Shrikumar, and A. Kundaje. Towards more realistic simulated datasets for benchmarking deep learning models in regulatory genomics. *BioRxiv*, dec 2021.

[37] A. R. Quinlan and I. M. Hall. BEDTools: a flexible suite of utilities for comparing genomic features. *Bioinformatics (Oxford, England)*, 26(6):841–2, mar 2010.

[38] I. Rauluseviciute, R. Riudavets-Puig, R. Blanc-Mathieu, J. A. Castro-Mondragon, K. Ferenc, V. Kumar, R. B. Lemma, J. Lucas, J. Chèneby, D. Baranasic, A. Khan, O. Fornes, S. Gundersen, M. Johansen, E. Hovig, B. Lenhard, A. Sandelin, W. W. Wasserman, F. Parcy, and A. Mathelier. JASPAR 2024: 20th anniversary of the open-access database of transcription factor binding profiles. *Nucleic Acids Research*, 52(D1):D174–D182, jan 2024.

[39] Y. Schiff, C.-H. Kao, A. Gokaslan, T. Dao, A. Gu, and V. Kuleshov. Caduceus: Bi-directional equivariant long-range DNA sequence modeling. *arXiv*, 2024.

[40] L. Song and G. E. Crawford. DNase-seq: a high-resolution technique for mapping active gene regulatory elements across the genome from mammalian cells. *Cold Spring Harbor Protocols*, 2010(2):pdb.prot5384, feb 2010.

[41] Z. Tang and P. K. Koo. Evaluating the representational power of pre-trained DNA language models for regulatory genomics. *BioRxiv*, mar 2024.

[42] The UniProt Consortium. UniProt: the universal protein knowledgebase. *Nucleic Acids Research*, 45(D1):D158–D169, jan 2017.

[43] B. D. Umans, A. Battle, and Y. Gilad. Where are the disease-associated eQTLs? *Trends in Genetics*, 37(2):109–124, feb 2021.

[44] A. E. Vinogradov. DNA helix: the importance of being GC-rich. *Nucleic Acids Research*, 31(7):1838–1844, apr 2003.

[45] I. E. Vorontsov, I. A. Eliseeva, A. Zinkevich, M. Nikonov, S. Abramov, A. Boytsov, V. Kamenets, A. Kasianova, S. Kolmykov, I. S. Yevshin, A. Favorov, Y. A. Medvedeva, A. Jolma, F. Kolpakov, V. J. Makeev, and I. V. Kulakovskiy. HOCOMOCO in 2024: a rebuild of the curated collection of binding models for human and mouse transcription factors. *Nucleic Acids Research*, 52(D1):D154–D163, jan 2024.

[46] H. Vu and J. Ernst. Universal annotation of the human genome through integration of over a thousand epigenomic datasets. *Genome Biology*, 23(1):9, jan 2022.

[47] Z. Zhou, Y. Ji, W. Li, P. Dutta, R. Davuluri, and H. Liu. DNABERT-2: Efficient foundation model and benchmark for multi-species genome. *arXiv*, 2023.


# Appendices

## A Code and Data

All our data and models can be found at `https://www.synapse.org/DART_Eval_Benchmark` unless otherwise specified. Our code can be found at `https://github.com/kundajelab/DART-Eval`.

## B Extended Background

### B.1 The Genome and Non-Coding Regulation

DNA, present in every cell, stores the complete set of instructions essential for life. It consists of a chain of nucleotides—adenine (A), thymine (T), guanine (G), and cytosine (C)—whose specific sequences encode functional elements. While genes, which code for proteins, are the most recognized of these elements, they constitute only a fraction of the genome. The complete DNA sequence of an organism is referred to as its genome.

Within genes, *coding* sequences specify the amino acid composition of proteins. Through the processes of transcription and translation, nucleotide triplets (codons) in these sequences are translated into amino acids, which form the building blocks of proteins.

However, in humans, coding sequences account for just around 1.5% of the genome. The remaining 98.5% includes vast regions of *non-coding* DNA, some of which play essential roles in regulating when, where, and to what extent genes are expressed. In multicellular organisms, gene activation and suppression are highly context-specific, enabling a single genome to support the development of diverse cell types across tissues and organs, each responding dynamically to internal and external signals.

Among the non-coding regions, *regulatory elements* play a crucial role in controlling gene expression according to cellular context. Unlike coding regions that directly produce proteins, these regulatory sequences contain nucleotide patterns that interact with specific DNA-binding proteins known as transcription factors (TFs). These interactions can alter the 3D structure of DNA, recruiting the molecular machinery required to activate or repress nearby genes.

Understanding non-coding regulatory elements remains challenging due to their sparse, combinatorial, and context-dependent nature. DNA-binding proteins vary in presence and behavior across different cell types, making the syntax of non-coding regulatory elements highly cell-type-specific. In this way, each gene is regulated by an array of elements, such as promoters and enhancers, each with distinct properties. Promoters are located close to the transcription start site, directly adjacent to the genes they regulate, while enhancers can reside thousands of base pairs away yet still regulate gene expression.

In summary, although non-coding regulatory elements do not produce proteins, they govern the spatiotemporal patterns of gene expression, enabling the complex regulatory landscapes that underpin cellular diversity and adaptive responses in multicellular organisms.

### B.2 Deep learning models of DNA elements

In recent years, several deep learning models have been developed to learn representations of different classes of DNA elements and predict their context-specific properties and activity. These models generally fall into two categories: supervised models, which are explicitly trained to map DNA sequence to associated properties or experimental measurements of biochemical activity, and self-supervised models, which learn representations of DNA sequences without any labeled data.

Supervised deep learning models have shown impressive results in modeling various types of biological sequences. For example, they have been successfully used to predict RNA splicing, a key post-transcriptional regulatory process [21], to predict protein structure from amino acid sequences [22, 24] and to predict chromatin and transcriptional activity from regulatory sequences in

diverse cell types [7, 6, 35]. These models rely on labeled data to learn mappings from sequence to structure or functional activity.

In contrast, self-supervised learning has shown great success in training protein language models. These models capture the complex syntax of protein-coding sequences [27, 42] by training on massive protein sequence datasets without requiring explicit functional labels. Due to the high information density and conserved syntax of protein-coding DNA across species, these models have proven especially adept at learning generalized protein representations that can be fine-tuned for downstream applications such as prediction of structure, interactions and even functional properties.

Recently, self-supervised DNA language models (DNALMs) have emerged as a novel approach, extending beyond protein-coding sequences to learn representations of entire genomes [34, 13, 16, 47]. Unlike protein language models, DNALMs are trained to capture the syntax across all classes of DNA elements, including diverse types of non-coding functional elements that often encode comlex and context-dependent syntax. By modeling the full spectrum of genomic sequences, DNALMs aim to capture both coding and non-coding syntax, potentially serving as foundation models for a wide array of downstream prediction tasks, potentially reducing the need for training specialized models from scratch.

## C   Datasets

### C.1   ENCODE candidate *cis*-regulatory elements

This dataset consists of a set of approximately 2.3 million high-confidence regulatory regions as curated by the ENCODE consortium. These regions are mainly enhancers or promoters, and they are active in at least one of a wide variety of cell types. Candidate regions were first identified by integrating cell type-specific DNAse-seq chromatin accessibility data with ChIP-seq data for the H3K27ac and H3K4me3 histone marks, which are biochemical markers associated with enhancers and promoters respectively. The final set of regions, available online on the ENCODE project website, is capped at a maximum length of 350 bp. We specifically used the cCRE list produced as part of phase IV of ENCODE, which provides an over-two-fold increase in identified cCREs from phase III. This extensive dataset serves as an ideal benchmark for evaluating language models' ability to capture essential regulatory DNA features. The dataset was downloaded from `https://www.encodeproject.org/files/ENCFF420VPZ/`. All ENCODE data is available for unrestricted use.

### C.2   HOCOMOCO transcription factor binding motifs

Each transcription factor recognizes specific DNA sequence motifs. To evaluate the models' ability to identify regulatory sequence features, we analyzed each motif independently. Among available motif databases, HOCOMOCO is widely used in the research community. It compiles motifs derived from ChIP-seq and HT-SELEX data, which measure protein-DNA binding, and uses the ChIPMunk motif discovery method to generate motif sequences. Version 12 of HOCOMOCO provides position-weight matrices (PWMs) for 949 human transcription factors, encompassing 1,443 unique motifs when accounting for subtypes. Each PWM provides nucleotide probabilities at each motif position, from which we derive consensus sequences by selecting the most probable nucleotide per position. The HOCOMOCO database also groups transcription factors into families, facilitating higher-level analyses. The database was downloaded from `https://hocomoco12.autosome.org/final_bundle/hocomoco12/H12CORE/formatted_motifs/H12CORE_meme_format.meme`. HOCOMOCO data is available under the WTFPL license.

### C.3   ATAC-seq and DNase-seq Peaks

The peak sets are summarized in Table S1 and Table S2. The cell-type-specific peak sets, identified by DESeq2, can be visualized in Figure S1.

ATAC-seq peaks and DNase-seq peaks are defined as regions of high chromatin accessibility in the genome. These datasets were downloaded from ENCODE. The GM12878 ATAC-seq peaks were obtained from `ENCFF748UZH`. The H1ESC ATAC-seq peaks were obtained from [35]. The HEPG2 ATAC-seq peaks were obtained from `ENCSR291GJU`. The IMR90 ATAC-seq peaks were obtained from `ENCFF243NTP`. The K562 ATAC-seq peaks were obtained from `ENCFF333TAT`. Amongst the

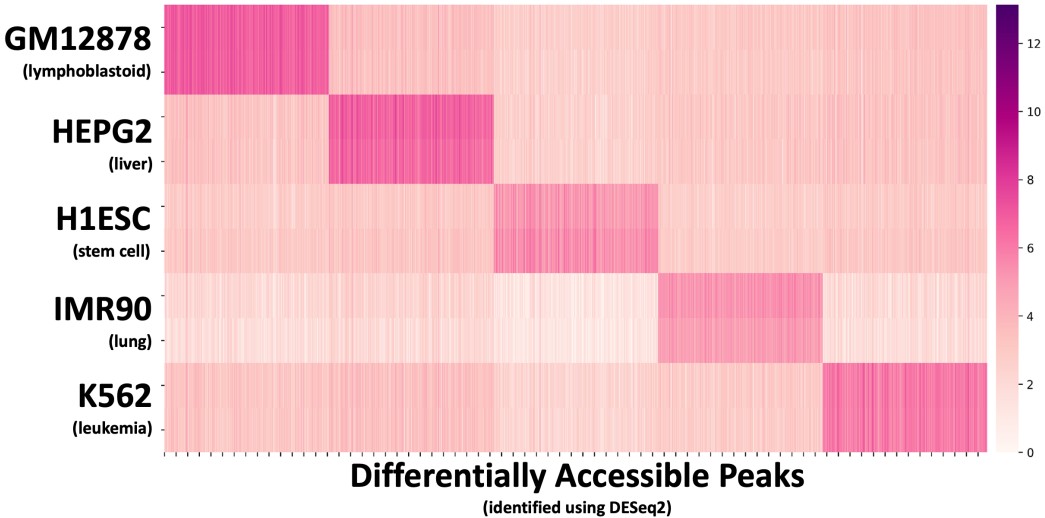

Figure S1: Correlations of per-motif embedding-based accuracies for each pair of models. Diagonals represent accuracy distribution for each model

Table S1: Overview of chromatin accessibility peak datasets used in training and evaluation: links to peaks

| Cell Type | ATAC-seq Peaks | DNase-seq Raw Files |
|---|---|---|
| GM12878 | ENCFF748UZH | ENCSR000EMT |
| H1ESC | [35] | ENCSR000EMU |
| HEPG2 | ENCSR291GJU | ENCSR149XIL |
| IMR90 | ENCFF243NTP | ENCSR477RTP |
| K562 | ENCFF333TAT | ENCSR000EOT |

Table S2: Overview of chromatin accessibility peak datasets used in training and evaluation: number of peaks

| Cell Type | # ATAC-seq Peaks | # DNase-seq Peaks | # DNase IDR Peaks | # Differentially Accessible Peaks |
|---|---|---|---|---|
| GM12878 | 277,999 | 127,079 | 70,897 | 45,184 |
| H1ESC | 104,250 | 103,000 | 48,188 | 49,208 |
| HEPG2 | 279,739 | 184,583 | 119,403 | 33,948 |
| IMR90 | 265,247 | 234,313 | 44,807 | 50,783 |
| K562 | 269,800 | 194,321 | 137,722 | 37,623 |

ATAC-seq datasets, there are a total of 277999 GM12878 peaks, 104250 H1ESC peaks, 279739 HEPG2 peaks, 265247 IMR90 peaks, and 269800 K562 peaks. All ENCODE data is available for unrestricted use.

The final set of DNase-seq peaks for all cell lines was obtained from [35]. The raw files were obtained from ENCODE and processed according to [35]. The GM12878 raw files were obtained from ENCSR000EMT. The H1ESC raw files were obtained from ENCSR000EMU. The HEPG2 raw files were obtained from ENCSR149XIL. The IMR90 raw files were obtained from ENCSR477RTP. The K562 raw files were obtained from ENCSR000EOT.

The final sets of high confidence, reproducible peaks for all cell lines were also obtained from [35].

### C.4 Variants that influence chromatin accessibility (caQTLs and dsQTLs)

Molecular quantitative trait loci (QTLs) are genetic variants that influence variation of a molecular activity (e.g. gene expression or chromatin accessibility) in a particular cell type across multiple individuals. DNase-seq QTLs (dsQTLs) are genetic variants associated with variation in chromatin accessibility, as measured by DNase-seq experiments. Chromatin accessibility QTLs (caQTLs)

Table S3: Cell-Type specific motifs in differentially accessible peaks

| | HOMER *-log10(q-value)* | | | | |
|---|---|---|---|---|---|
| Motif Name | GM12878 | H1ESC | HEPG2 | IMR90 | K562 |
| IRF1 | 10 | 0 | 0 | 0 | 0 |
| IRF2 | 10 | 0 | 0 | 0 | 0 |
| SpiB | 10 | 0 | 0 | 0 | 0 |
| Oct4 | 0 | 10 | 0 | 0 | 0 |
| Sox2 | 0 | 10 | 0 | 0 | 0 |
| Sox6 | 0 | 10 | 0 | 0 | 0 |
| Hnf4a | 0 | 0 | 10 | 0 | 0 |
| FoxA1 | 0 | 0 | 10 | 0 | 0 |
| Hnf1 | 0 | 0 | 10 | 0 | 0 |
| ATF3 | 0 | 0 | 0 | 10 | 0 |
| Fosl2 | 0 | 0 | 0 | 10 | 0 |
| Jun-AP1 | 0 | 0 | 0 | 10 | 0 |
| Gata1 | 0 | 0 | 0 | 0 | 10 |
| KLF4 | 0 | 0 | 0 | 0 | 10 |
| Gata2 | 0 | 0 | 0 | 0 | 10 |

Table S4: Overview of variant datasets

| Dataset Name | # Total Variants | # Significant Variants | # Control Variants | Original Source | Filtered Source |
|---|---|---|---|---|---|
| Chromatin QTLs in African LCLs | 219,382 | 6,821 | 77,999 | [15] | syn59449898 |
| DNase QTLs in Yoruban LCLs | 28,309 | 560 | 26,813 | [14] | syn59449898 |

are genetic variants associated with variation in chromatin accessibility measured using ATAC-seq experiments. Genomic elements with strong ATAC-seq or DNase-seq signal are typically regulatory elements bound by TFs. We used two QTL datasets to evaluate all the models (Table S4). We downloaded the processed CaQTLs from [35] (File `variant_effect_benchmarking.tsv.gz` from Synapse repository `syn59449898`).

*caQTLs in African LCLs.* The first dataset consists of 219,382 variants and their effect sizes and statistical significance of association with variation of ATAC-seq signal across 100 lymphoblastoid cell-lines from individuals 6 African ancestry subpopulations (ESN, GWD, LWK, MSL, YRI, and MKK) [15]. After filtering the variants using the procedure described in [35], we were left with 77,999 control variants and 6,821 statistically significant caQTLs. Variants are restricted to fall within ATAC-seq peaks identified in the entire cohort in order to enrich for likely causal caQTLs. The data is available under the Creative Commons Attribution 4.0 International License.

*DNase QTLs in African LCLs.* We obtained a dataset from [14], which comprises 560 statistically significanct DNase I sensitivity QTL (dsQTL) variants and 26,813 control variants. We filtered the variants using the procedure described in [35]. Variants are restricted to fall within DNase-seq peaks identified in the entire cohort in order to enrich for likely causal caQTLs

# D Models

## D.1 Zero-Shot Model Evaluations

All pre-trained models used in this study were obtained from HuggingFace using the documentation provided in each model's README.

For all models, sequence embeddings were derived from the output of the last hidden layer when performing inference on the input sequence. Embeddings for auxiliary tokens like `<CLS>`, `<start>`, and `<end>` were removed, and the remaining embeddings were averaged to produce an overall sequence representation. For models using byte-pair encodings, where tokens represent variable numbers of nucleotides, this average is weighted by the number of nucleotides in each token. This embedding process is used in all embedding comparison tasks in this study.

To calculate model (pseudo-)likelihoods for an input sequence, obtain the predicted logits for each token. For autoregressive models, this can be done with a single forward pass, where each token is conditioned on preceding tokens. For masked models, we successively masked each token and compute predicted logits at the masked position conditioned on all other tokens. Unscaled logits

were then converted into log-likelihoods using log softmax, and the log-likelihood for the true token choice at each position is isolated. These token-level log-likelihoods were then summed across tokens (multiplied in log space) to produce the overall sequence likelihood. This sequence-level log-likelihood methodology was used for all likelihood-based comparisons in this study.

## D.2 Probed and Fine-Tuned Models

For final-layer probing, the base pre-trained model weights were frozen. Outputs from the final hidden layer were passed to an additional CNN-based probing head. Embeddings were converted from token space to sequence space by repeating each token embedding by the number of nucleotides spanned by the token, as in [30] and [41]. The probing head consists of a linear projection to 32 dimensions, two convolutional layers of width 8 and 32 filters, a sum pooling layer, and a linear layer to produce the final output. ReLU activations are applied after each intermediate layer. Probing heads were trained using Adam with a learning rate of $2e^{-3}$.

Fine-tuning utilized LoRA, a widely-used parameter-efficient fine-tuning method that performs low-rank updates to model parameters [20]. For consistency across multiple architectures, we applied fine-tuning to all linear and convolutional layers. We used each model's included classifier head, trained from scratch. LoRA parameters included a rank of 8, an $\alpha$ of 16, and a dropout of 0.05. Optimization used AdamW with a learning rate of $1e^{-4}$ and a weight decay of 0.01.

We used a consistent train, validation, and test split across all experiments, at an approximate 4:1 train and validation to test split, and an approximate 9:1 train to validation split. Our test set consists of chromosomes 5, 10, 14, 18, 20, 22. Our validation set consists of chromosomes 6 and 21, and our training set consists of all other chromosomes. For all models, we evaluated the checkpoint with the lowest validation loss. All reported numbers were computed on the test set unless otherwise stated.

## D.3 *Ab initio* Models

For the chromatin accessibility regression models - which were also used in the variant interpretation task - our *Ab initio* baseline was ChromBPNet, a convolutional neural network that can predict the magnitude and shape of chromatin accessibility profiles at base-pair resolution from an input DNA sequence. ChromBPNet takes as input a one-hot encoded DNA sequence of length 2,114, passing it through a single convolutional layer followed by 8 dilated residual layers of increasing kernel size. The output of these layers is used to make two predictions. First, a Global Average Pooling (GAP) layer is applied, followed by a linear layer to predict the total ATAC-seq or DNase-seq read counts within the central 1,000 bp of the input. Only this prediction was used to compare with the probed and fine-tuned language models. Second, the convolutional output is passed through another convolutional layer with a large kernel and only one channel, producing a predicted base-level probability profile of reads over the output region. By multiplying both model outputs together, one can obtain the predicted read counts at each position in the output region. The count prediction was trained using mean squared error loss, while the profile head was trained using log-likelihood loss based on a multinomial distribution. Separate ChromBPNet models are trained on each chromatin accessibility dataset. We utilized already trained ChromBPNet models from the ENCODE project for each dataset in this study.

For all tasks except chromatin accessibility regression and variant effect prediction, we compared against a small custom-trained CNN resembling the probing head we use, as it has a similar model capacity. This model consists of two parts: an embedding block - designed to produce simple sequence embeddings of similar dimensionality to DNALM embeddings - followed by an output head. The architecture of the output head is identical to the head used for probing. The embedding block takes in a one-hot encoded DNA sequence as input and applies a single convolutional layer of width 41 and 256 channels. This output is summed with a learned single-channel positional embedding, up-projected to 256 channels. The resulting embeddings then serve as the input to the output head. Models were trained using Adam with a learning rate of $1e^{-3}$.

For the cell type-specific regulatory DNA task, we implemented an additional larger *Ab initio* baseline resembling the ChromBPNet architecture. Differences from ChromBPNet are (1) 7 dilated residual convolutional layers instead of 8, (2) removal of the base-pair-resolution prediction head, and (3) the addition of a single-channel learned positional encoding, incorporated after the initial convolutional layer. Models were trained using Adam with a learning rate of $1e^{-4}$.

Train, validation, and test folds are identical to those used for fine-tuning and probing.

# E  Tasks and Results

Model training and evaluations were performed on Kundaje Lab machines, the Stanford Sherlock HPC cluster, and Google Cloud VMs. We utilized a combination of NVIDIA L40S, A100 (40 and 80 GB), V100, and Titan X GPUs depending on availability.

## E.1  Distinguishing regulatory DNA from background sequences

Our first task tests whether models could discriminate regulatory elements from synthetic background sequences. For our positive set of regulatory elements, we used the ENCODE cCRE list of approximately 2.3 million high-confidence regulatory regions. We then performed dinucleotide shuffling on each cCRE sequence to produce a matched set of synthetic negative background sequences, in which negative sequences retain the same sequence composition as their positive counterparts but lack the binding motifs that promote activity. To ensure reproducibility of the shuffling process, the algorithm was seeded by the SHA-256 hash of the input region's genomic coordinates.

We then tested the models' binary classification performance in zero-shot, probed, and fine-tuned settings. In the zero-shot setting, we calculated the likelihood for each cCRE and background sequence, with a correct prediction defined as a higher likelihood for a cCRE than its corresponding background sequence. For both the probing and fine-tuned settings, we trained classifiers to predict which category a sequence belongs to.

For the zero-shot evaluation, performance metrics included accuracy and a one-sided Wilcoxon Rank-Sum Test between the cCRE and control likelihoods. For the other settings, metrics included accuracy, AUROC, and AUPRC.

## E.2  Assessing sensitivity to known regulatory sequence motifs

Models were then evaluated for their ability to recognize individual transcription factor binding motifs. We used a list of 1,443 consensus transcription factor (TF) motif sequences from the HOCOMOCO v12 database. 100 neutral background sequences were randomly chosen from the cCRE classification task background set. Specifically, for each combination of neutral sequence and motif, the following sequences were considered:

1. Neutral: the original neutral sequence
2. Positive: the neutral sequence with the motif inserted at the center (for a length-$n$ motif, the central $n$ nucleotides of the sequence were replaced with the motif)
3. Negative: the control sequence with a shuffled version of the motif inserted at the center
4. Reverse complement of the neutral (1)
5. Reverse complement of the positive (2)
6. Reverse complement of the negative (3)

Taken together, this procedure resulted in a dataset of 577,400 unique sequences.

We employed likelihood and embedding-based approaches for this task. For the likelihood approach, we determined whether the predicted likelihood was higher for each positive sequence than for each corresponding negative sequence. 200 such pairs exist in the dataset for each motif, and we defined a model's accuracy for that motif as the proportion of pairs where the positive sequence had a higher predicted likelihood. We also utilized the results to compute a one-sided Wilcoxon Rank Sum significance test for each motif. Note that neutral sequences were not used for this analysis.

We also evaluated using an embedding-based approach with the following procedure:

1. Let $s_\square$ be a raw or reverse-complemented neutral sequence. Let $s_+$ be the corresponding positive sequence, and let $s_-$ be the corresponding negative sequence.
2. We calculate $d_+$, the embedding distance between $s_+$ and $s_\square$, and $d_-$, the embedding distance between $s_-$ and $s_\square$. Cosine distance is used as the embedding distance metric.

3. The prediction for the triplet $(s_+, s_-, s_\square)$ is considered correct if $d_+ > d_-$.

As with the likelihood evaluation, 200 such pairs exist per motif, allowing us to obtain an accuracy metric for each motif. Also as before, we evaluated significance using a one-sided Wilcoxon Rank Sum Test.

We additionally used metadata from the HOCOMOCO database to group motifs into motif families. We then aggregated per-motif accuracy metrics to the family level.

### E.3 Learning cell-type-specific regulatory sequence features

We next evaluated whether models can discriminate accessible regulatory regions in different cell types that possess distinct sets of active sequence features. We utilized ATAC-seq peaks from five cell lines: GM12878, H1ESC, HEPG2, IMR90, and K562, with multiple biological replicates for each cell type. Details are in Appendix C.3. These cell lines are extensively studied and are also known to differ in the set of key transcription factors that regulate accessibility in each cell-line. We identified differentially-active peak sequences using DESeq2, a negative-binomial-model-derived statistical test for read-count-based experimental assays. Specifically, we formed a consensus peak set by merging and deduplicating peaks from each cell type. Then, we counted the number of ATAC reads intersecting each consensus peak region in each cell type. Then, we used DESeq2 in a one-vs-others fashion for each cell type, where the positive class corresponds to $C_i$, the cell type for which we are finding the differential peaks, and the negative set $= \{C_j\}$ with $j \neq i$ corresponding to all the other cell types. Our final differential peak sets were chosen with a positive log fold change $> 1$ and an adjusted $p$-value $< 0.001$. We only kept peaks with differential activity in exactly one cell type. We summarized the number of differentially accessible peaks in each cell type in Table S2. We validated our differential peak set using Homer [19]. HOMER is a *de novo* motif discovery algorithm that scores motifs by looking for motifs with differential enrichment between two sets of sequences. For our purposes, we used the differentially accessible peak set in one cell type as the target set and the differentially accessible peak sets in all other cell types as the background set, and we repeated this for all cell types. HOMER takes the motifs identified from the *de novo* motif discovery step and compares them against a library of known motifs in JASPAR [17]. In Table S3, we present the negative log of the Benjamini-Hochberg-adjusted $q$ values from the HOMER motif discovery, with $-log(q)$ capped at 10.

In the zero-shot setting, we further restricted the peak sets to the top 5000 differential peaks per cell line, based on the adjusted DESeq2 $p$-value. On these peak sets, we produced model embeddings for each peak sequence. For the baseline, we computed motif scores using FIMO [18], which scans a collection of DNA sequences for occurrences of one or motifs from the HOCOMOCO database described in Appendix C.2. We intersected the motif hits with the peaks using BedTools [37] and constructed bag-of-motifs embeddings for each peak where each entry is the sum of the $-\log_{10}(\text{FIMO } q\text{-value})$ for a particular motif in that peak sequence. We then selected for the most variable motifs using a permutation method comparing the sum of the motif across all the peaks in each subsampled peak set. We performed the subsampling procedure 1000 times with each subsampled peak set consisting of 100 peaks. (Note that ground-truth labels are not used at any stage when constructing baseline embeddings.) We then performed $k$-means clusterings on each set of embeddings, with $k$ set to 50. The ability of the clustering to differentiate peaks from different cell lines was quantified through the adjusted Mutual Information Score between the cluster labels and the true cell line labels for each peak. The Adjusted Mutual Information (AMI) score, a common method to evaluate clustering results, measures concordance between two sets of labels. Its maximum value is 1.0, with values close to 0 indicating random labeling and values close to 1 indicating a perfect match between clusters and labels. We obtained AMI scores from 100 different k-means clustering runs and define a conservative 95% confidence interval around the mean as the difference between the mean and the 2.5% quantile or the 97.5% quantile, whichever is greater.

In the probing and fine-tuning settings, we trained a five-way classifier to predict the cell line from which each peak was derived. Important metrics included accuracy, AUROC, and AUPRC.

### E.4 Predicting quantitative measures of regulatory activity from sequence

This task involves predicting quantitative measurements of chromatin accessibility from sequence, quantified as DNase-Seq read counts over the sequence. DNase-Seq peaks (regions of high accessi-

bility) and count data were obtained from the ENCODE consortium for the same set of 5 cell lines used in earlier tasks: GM12878, H1ESC, HEPG2, IMR90, and K562.

ChromBPNet models were trained on the same data and used as our baseline models. We utilized the same training setup for our probing and fine-tuning models so that inputs and labels were identical to those for ChromBPNet. Specifically, the ChromBPNet preprocessing pipeline involved filtering peaks to remove read count outliers and then expanding the remaining peaks to size 2,114. In addition to accessibility peaks, ChromBPNet is also trained on matched negative genomic background sequences. Specifically, for each peak, a negative region was selected from elsewhere in the genome with the same GC content but does not fall within the peak set. The ratio of peaks to negatives in each training batch is 10:1. Within batches, half the sequences were reverse-complemented, and each sequence was shifted a maximum of 500bp in either direction, to ensure the area of highest accessibility is not always at the center of the input. The ground-truth activity for a given input sequence was defined as the number of read endpoints intersecting the central 1,000 bp.

Quantitative predictions were evaluated using the Pearson and Spearman (rank-normalized) correlation between the predicted accessibility and measured accessibility. Metrics were computed across peaks only and also across peaks and background sequences. Models were also evaluated based on their ability to classify peaks from background sequences, quantified by AUROC and AUPRC. For classification metrics, the set of positives was restricted to high-confidence, reproducible peaks, identified using the Irreproducing Discovery Rate (IDR) method [25] that determines whether peaks identified in replicate experiments are rank consistent and reproducible.

### E.5   Predicting counterfactual effects of regulatory genetic variants

A critical challenge in human genetics is predicting how genetic variants affect gene regulation through changes in chromatin accessibility. Models trained to predict regulatory activity from sequence (S2A models) (such as those in Section4.4) are typically used in a counterfactual setting to predict the effects of genetic variants on regulatory activity. This is a particularly challenging task since the S2A models are never directly trained on genetic variation data. We evaluated the ability of DNALMs to prioritize and predict the quantitative effects of regulatory genetic variants that impact chromatin accessibility.

Each variant is a single nucleotide polymorphism (SNP) consisting of a pair of alleles, a reference allele $x_{ref} \in \{$A,C,G,T$\}$ and an alternate allele $x_{alt} \in \{$A,C,G,T$\}$, together with a label $y \in \{$1,0$\}$, indicating whether the variant is a statistically significant chromatin accessibility QTL (dsQTL or caQTL) or a background variant. All genomic variant coordinates for the caQTL dataset are based on the human reference genome version GRCh38, whereas variant coordinates for the dsQTL dataset are based on the human reference genome version GRCh37.

Each allele of a variant was scored by taking a sequence of length 2114, where the variant allele was placed in the center of a 2114-length sequence, with the remaining sequence provided as context. Both sequences, with reference and alternate alleles respectively, were passed through the model to obtain scores for each.

In the zero-shot embedding setting, given reference and alternate alleles, two embeddings were computed, and the cosine distance between the embeddings was used as the allelic effect score of the variant. In the zero-shot likelihood setting, the variant position was masked out and the likelihoods at the mask token with respect to the reference and alternate alleles are compared. In supervised settings, we evaluated the predicted counts log fold change between the two alleles.

Table S5: Resource requirements of evaluated DNALMs.

| Model | Variant | Parameters | Inference | | Training | |
|---|---|---|---|---|---|---|
| | | | Runtime (ms) | Memory (GB) | Runtime (ms) | Memory (GB) |
| Caduceus | `ps_131k_d-256_n-16` | 7,725,568 | 239.67 ± 1.11 | 1.07 ± 0.00 | 834.78 ± 3.67 | 40.82 ± 0.00 |
| DNABERT-2 | `117M` | 117,069,313 | 104.17 ± 2.06 | 1.59 ± 0.03 | 325.93 ± 6.74 | 8.81 ± 0.17 |
| GENA-LM | `bert-large-t2t` | 336,658,433 | 194.04 ± 5.47 | 3.17 ± 0.02 | 502.53 ± 13.54 | 18.35 ± 0.73 |
| HyenaDNA | `large-1m` | 6,550,784 | 59.51 ± 0.57 | 0.94 ± 0.00 | 174.08 ± 3.77 | 7.57 ± 0.00 |
| Mistral-DNA | `v1-1.6B-hg38` | 1,607,677,440 | 129.63 ± 7.41 | 9.35 ± 0.03 | 351.36 ± 13.54 | 14.69 ± 0.24 |
| Nucleotide Transformer | `v2-500m-multi-species` | 494,134,738 | 289.58 ± 0.76 | 4.44 ± 0.00 | 733.30 ± 2.12 | 22.21 ± 0.00 |

DNALM resource requirements per batch of 64 sequences of length 2114 bp. Statistics are displayed as mean ± standard deviation. Values include each model's classification head. Gradients were computed for all model parameters when measuring training resource requirements. This evaluation was conducted on an Nvidia L40S GPU.

Table S6: Regulatory element identification extended results

| Setting | Model | Absolute Accuracy | Paired Accuracy | AUROC | AUPRC |
|---|---|---|---|---|---|
| Probed | Caduceus | 0.7257 ± 4.0344e-04 | 0.8961 ± 2.7588e-04 | 0.8203 | 0.8319 |
| | DNABERT-2 | 0.8467 ± 3.2579e-04 | 0.9428 ± 2.1003e-04 | 0.9314 | 0.9366 |
| | GENA-LM | 0.8867 ± 2.8661e-04 | 0.9594 ± 1.7857e-04 | 0.9580 | 0.9627 |
| | HyenaDNA | 0.8475 ± 3.2511e-04 | 0.9347 ± 2.2338e-04 | 0.9274 | 0.9300 |
| | Mistral-DNA | 0.7591 ± 3.8671e-04 | 0.8587 ± 3.1503e-04 | 0.8430 | 0.8492 |
| | Nucleotide Transformer | 0.8194 ± 3.4785e-04 | 0.9168 ± 2.4980e-04 | 0.9025 | 0.9043 |
| Fine-Tuned | Caduceus | 0.9030 ± 2.6769e-04 | 0.9707 ± 1.5260e-04 | 0.9723 | 0.9746 |
| | DNABERT-2 | 0.9131 ± 2.5467e-04 | 0.9730 ± 1.4650e-04 | 0.9745 | 0.9769 |
| | GENA-LM | 0.9095 ± 2.5946e-04 | 0.9722 ± 1.4875e-04 | 0.9746 | 0.9772 |
| | HyenaDNA | 0.8768 ± 2.9721e-04 | 0.9523 ± 1.9264e-04 | 0.9505 | 0.9530 |
| | Mistral-DNA | 0.8167 ± 3.4986e-04 | 0.9053 ± 2.6476e-04 | 0.9017 | 0.9068 |
| | Nucleotide Transformer | **0.9200 ± 2.4530e-04** | **0.9762 ± 1.3775e-04** | **0.9781** | **0.9804** |
| *Ab initio* | Probing-head-like | 0.8460 ± 3.2640e-04 | 0.9320 ± 2.2765e-04 | 0.927 | 0.931 |

Table S7: Quantiles of motif identification accuracies for each model

| Setting | Model | 0% | 25% | 50% | 75% | 100% |
|---------|-------|-----|-----|-----|-----|------|
| Likelihood | Caduceus | 0.035 | 0.420 | 0.570 | 0.700 | **1.000** |
| | DNABERT-2 | 0.145 | **0.495** | 0.590 | 0.685 | **1.000** |
| | GENA-LM | 0.055 | 0.475 | 0.620 | 0.740 | **1.000** |
| | HyenaDNA | 0.000 | 0.420 | **0.645** | **0.820** | 0.995 |
| | Mistral-DNA | 0.002 | 0.455 | 0.625 | 0.770 | **1.000** |
| | Nucleotide Transformer | 0.200 | 0.465 | 0.565 | 0.658 | 0.995 |
| Embedding | Caduceus | 0.370 | 0.475 | 0.500 | 0.525 | 0.630 |
| | DNABERT-2 | 0.375 | 0.480 | 0.500 | 0.525 | 0.635 |
| | GENA-LM | **0.390** | 0.485 | 0.510 | 0.535 | 0.630 |
| | HyenaDNA | 0.370 | 0.480 | 0.505 | 0.530 | 0.610 |
| | Mistral-DNA | **0.390** | 0.475 | 0.495 | 0.520 | 0.635 |
| | Nucleotide Transformer | **0.390** | 0.480 | 0.505 | 0.530 | 0.615 |

Pairwise Comparisons of Likelihood-Based Motif Identification Performance

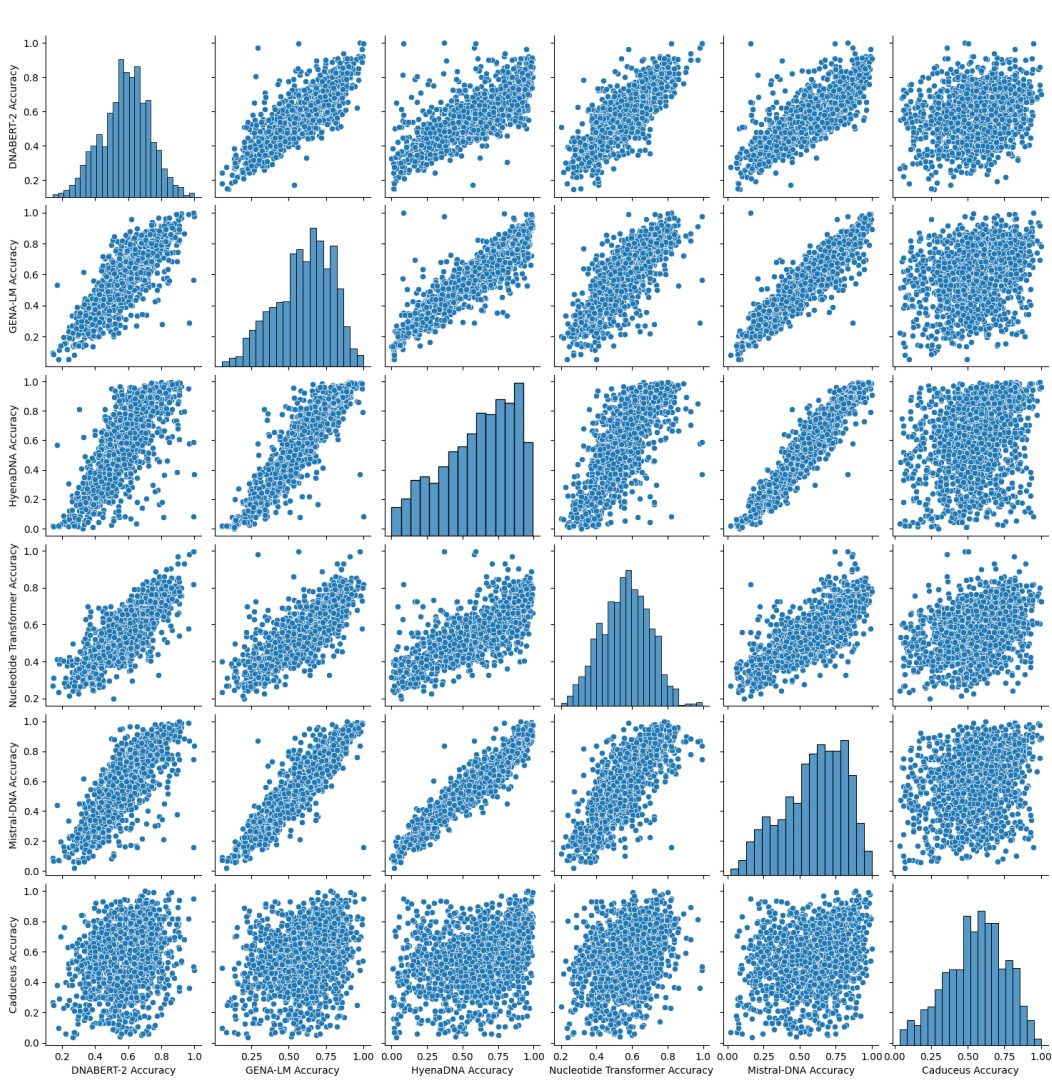

Figure S2: Correlations of per-motif likelihood-based accuracies for each pair of models. Diagonals represent accuracy distribution for each model

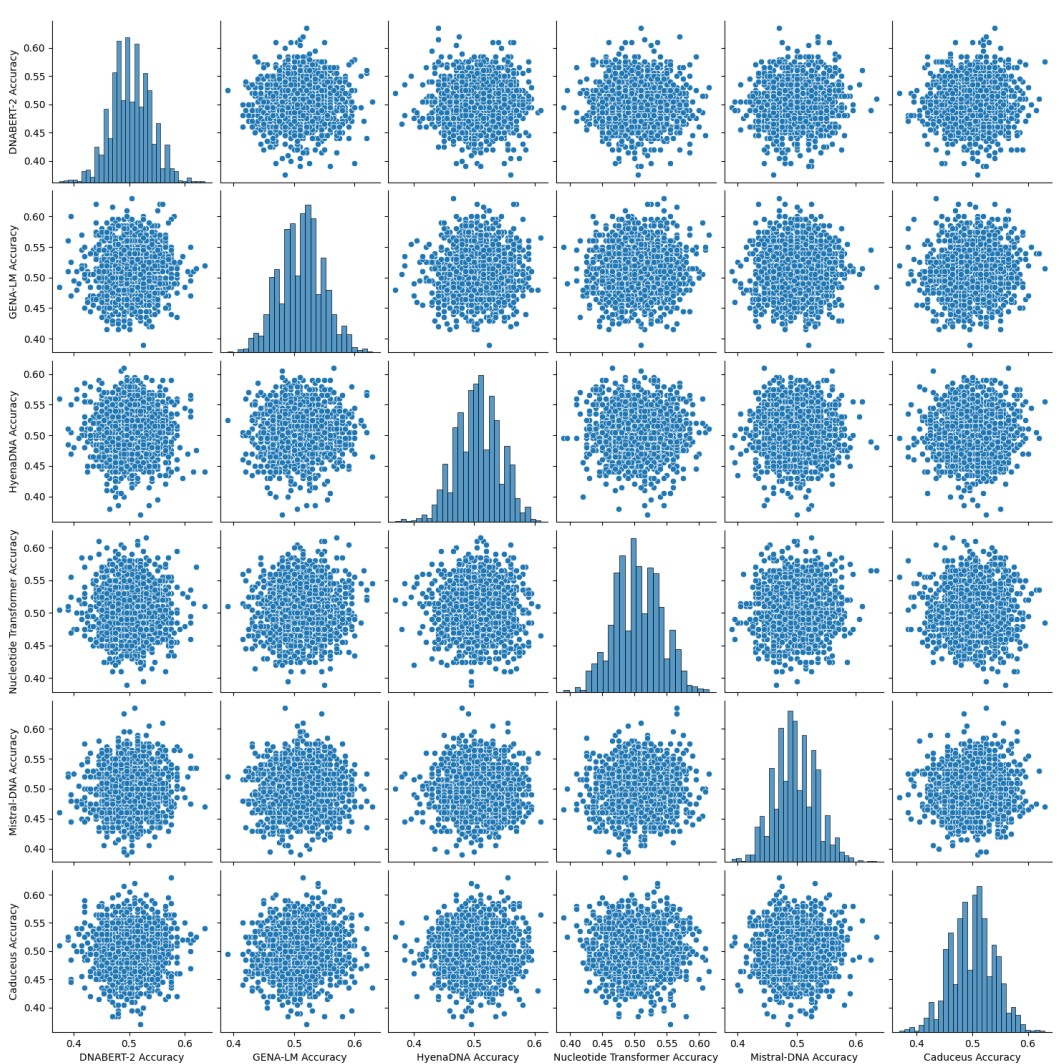

Figure S3: Correlations of per-motif embedding-based accuracies for each pair of models. Diagonals represent accuracy distribution for each model

## DNABERT-2

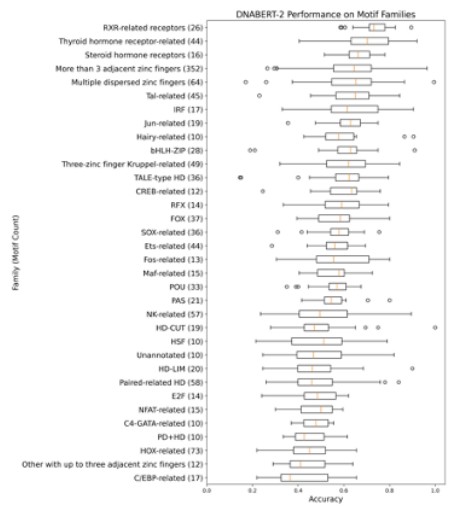

## GENA-LM

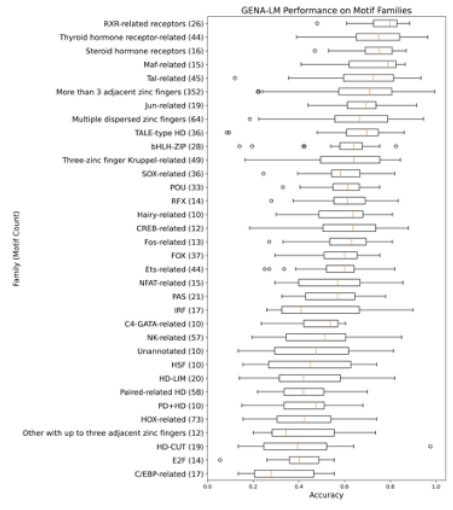

## HyenaDNA

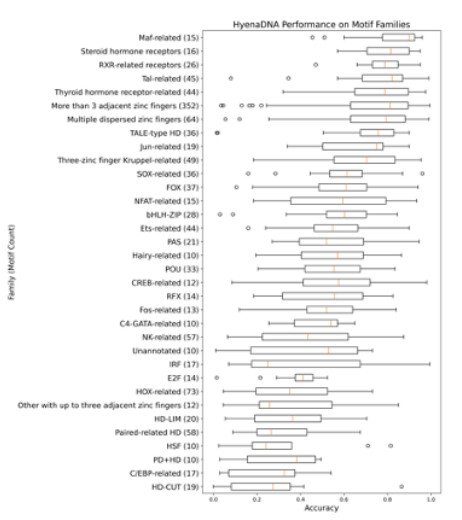

## Nucleotide Transformer

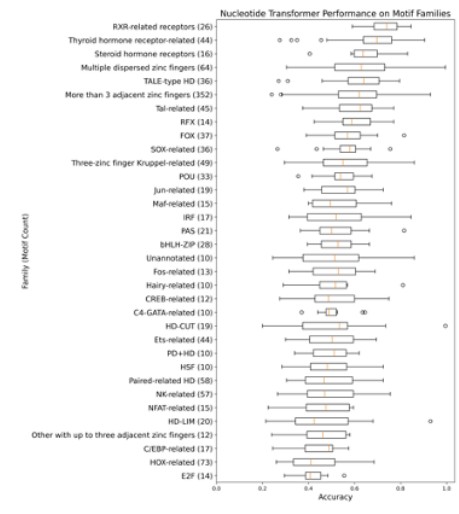

## Mistral-DNA

## Caduceus

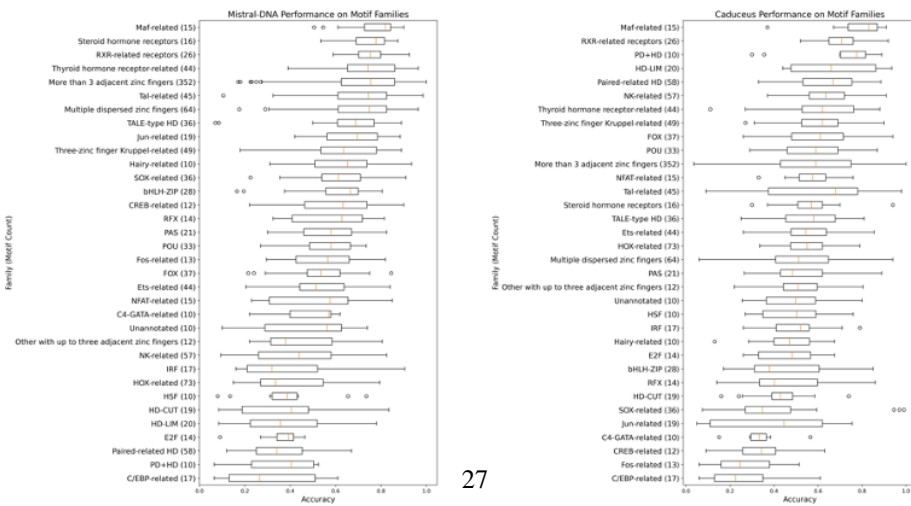

Figure S4: Likelihood-based motif detection accuracy distributions for each motif family

**DNABERT-2**

**GENA-LM**

**HyenaDNA**

**Nucleotide Transformer**

**Mistral-DNA**

**Caduceus**

Figure S5: Embedding-based motif detection accuracy distributions for each motif family

Table S8: Cell-type specific element classification results (multi-class overall accuracy)

| Setting | Model | Overall Accuracy |
|---|---|---|
| Probed | Caduceus | 0.281 ± 1.893e-03 |
| | DNABERT-2 | 0.371 ± 2.033e-03 |
| | GENA-LM | 0.383 ± 2.046e-03 |
| | HyenaDNA | 0.587 ± 2.073e-03 |
| | Mistral-DNA | 0.329 ± 1.979e-03 |
| | Nucleotide Transformer | 0.420 ± 2.078e-03 |
| Fine-Tuned | Caduceus | **0.671 ± 1.978e-03** |
| | DNABERT-2 | 0.650 ± 2.008e-03 |
| | GENA-LM | 0.636 ± 2.025e-03 |
| | HyenaDNA | 0.610 ± 2.053e-03 |
| | Mistral-DNA | 0.402 ± 2.064e-03 |
| | Nucleotide Transformer | 0.632 ± 2.030e-03 |
| *Ab initio* | ChromBPNet-like | 0.667 ± 1.984e-03 |
| | Probing-head-like | 0.474 ± 2.102e-03 |

Table S9: Cell-type specific element classification results (GM12878 vs. rest)

| Setting | Model | Accuracy | AUROC | AUPRC |
|---|---|---|---|---|
| Probed | Caduceus | 0.7912 ± 1.7110e-03 | 0.5354 | 0.2300 |
| | DNABERT-2 | 0.7891 ± 1.7173e-03 | 0.6516 | 0.3225 |
| | GENA-LM | 0.7919 ± 1.7089e-03 | 0.6267 | 0.2949 |
| | HyenaDNA | 0.8556 ± 1.4798e-03 | 0.8494 | 0.6799 |
| | Mistral-DNA | 0.7912 ± 1.7110e-03 | 0.5822 | 0.2745 |
| | Nucleotide Transformer | 0.8122 ± 1.6440e-03 | 0.7440 | 0.4857 |
| Fine-Tuned | Caduceus | 0.8854 ± 1.3411e-03 | 0.8998 | 0.7839 |
| | DNABERT-2 | 0.8784 ± 1.3761e-03 | 0.8939 | 0.7654 |
| | GENA-LM | 0.8687 ± 1.4219e-03 | 0.8770 | 0.7304 |
| | HyenaDNA | 0.8662 ± 1.4332e-03 | 0.8755 | 0.7226 |
| | Mistral-DNA | 0.7966 ± 1.6947e-03 | 0.6871 | 0.3977 |
| | Nucleotide Transformer | 0.8680 ± 1.4251e-03 | 0.8800 | 0.7337 |
| *Ab initio* | Probing-head-like | 0.8120 ± 1.6449e-03 | 0.7538 | 0.5367 |
| | ChromBPNet-like | **0.8865 ± 1.3355e-03** | **0.9026** | **0.7889** |

Table S10: Cell-type specific element classification results (H1ESC vs. rest)

| Setting | Model | Accuracy | AUROC | AUPRC |
|---|---|---|---|---|
| Probed | Caduceus | 0.7775 ± 1.7510e-03 | 0.6221 | 0.2889 |
| | DNABERT-2 | 0.7907 ± 1.7126e-03 | 0.7572 | 0.4755 |
| | GENA-LM | 0.8062 ± 1.6640e-03 | 0.7868 | 0.5473 |
| | HyenaDNA | 0.8448 ± 1.5243e-03 | 0.8893 | 0.7312 |
| | Mistral-DNA | 0.7777 ± 1.7503e-03 | 0.6775 | 0.3497 |
| | Nucleotide Transformer | 0.7962 ± 1.6958e-03 | 0.7953 | 0.5251 |
| Fine-Tuned | Caduceus | **0.8941 ± 1.2956e-03** | **0.9370** | **0.8353** |
| | DNABERT-2 | 0.8861 ± 1.3373e-03 | 0.9300 | 0.8163 |
| | GENA-LM | 0.8806 ± 1.3653e-03 | 0.9229 | 0.7977 |
| | HyenaDNA | 0.8684 ± 1.4234e-03 | 0.9060 | 0.7612 |
| | Mistral-DNA | 0.7776 ± 1.7507e-03 | 0.7623 | 0.4759 |
| | Nucleotide Transformer | 0.8811 ± 1.3628e-03 | 0.9252 | 0.8054 |
| *Ab initio* | Probing-head-like | 0.8289 ± 1.5855e-03 | 0.8360 | 0.6458 |
| | ChromBPNet-like | 0.8856 ± 1.3401e-03 | 0.9286 | 0.8230 |

Table S11: Cell-type specific element classification results (HEPG2 vs. rest)

| Setting | Model | Accuracy | AUROC | AUPRC |
|---|---|---|---|---|
| Probed | Caduceus | 0.8167 ± 1.6290e-03 | 0.6801 | 0.3466 |
| | DNABERT-2 | 0.8270 ± 1.5925e-03 | 0.7619 | 0.4490 |
| | GENA-LM | 0.8266 ± 1.5939e-03 | 0.7727 | 0.4608 |
| | HyenaDNA | 0.8586 ± 1.4668e-03 | 0.8620 | 0.6331 |
| | Mistral-DNA | 0.8165 ± 1.6297e-03 | 0.7235 | 0.4038 |
| | Nucleotide Transformer | 0.8230 ± 1.6067e-03 | 0.7831 | 0.4672 |
| Fine-Tuned | Caduceus | **0.8823 ± 1.3567e-03** | **0.9009** | **0.7349** |
| | DNABERT-2 | 0.8750 ± 1.3923e-03 | 0.8910 | 0.7026 |
| | GENA-LM | 0.8745 ± 1.3946e-03 | 0.8871 | 0.6956 |
| | HyenaDNA | 0.8623 ± 1.4508e-03 | 0.8737 | 0.6539 |
| | Mistral-DNA | 0.8225 ± 1.6084e-03 | 0.7344 | 0.4121 |
| | Nucleotide Transformer | 0.8706 ± 1.4129e-03 | 0.8811 | 0.6797 |
| *Ab initio* | Probing-head-like | 0.8223 ± 1.6092e-03 | 0.7574 | 0.4163 |
| | ChromBPNet-like | 0.8739 ± 1.3975e-03 | 0.8938 | 0.7062 |

Table S12: Cell-type specific element classification results (IMR90 vs. rest)

| Setting | Model | Accuracy | AUROC | AUPRC |
|---|---|---|---|---|
| Probed | Caduceus | 0.7613 ± 1.7945e-03 | 0.5765 | 0.2806 |
| | DNABERT-2 | 0.7618 ± 1.7934e-03 | 0.6907 | 0.3901 |
| | GENA-LM | 0.7629 ± 1.7905e-03 | 0.7137 | 0.4250 |
| | HyenaDNA | 0.8377 ± 1.5525e-03 | 0.8816 | 0.7216 |
| | Mistral-DNA | 0.7612 ± 1.7949e-03 | 0.6428 | 0.3469 |
| | Nucleotide Transformer | 0.7759 ± 1.7554e-03 | 0.7794 | 0.5197 |
| Fine-Tuned | Caduceus | **0.8784 ± 1.3759e-03** | **0.9294** | **0.8211** |
| | DNABERT-2 | 0.8626 ± 1.4492e-03 | 0.9220 | 0.7993 |
| | GENA-LM | 0.8582 ± 1.4685e-03 | 0.9112 | 0.7811 |
| | HyenaDNA | 0.8531 ± 1.4901e-03 | 0.9075 | 0.7729 |
| | Mistral-DNA | 0.7775 ± 1.7511e-03 | 0.7479 | 0.4938 |
| | Nucleotide Transformer | 0.8586 ± 1.4669e-03 | 0.9204 | 0.7977 |
| *Ab initio* | Probing-head-like | 0.8049 ± 1.6683e-03 | 0.8065 | 0.5968 |
| | ChromBPNet-like | 0.8744 ± 1.3951e-03 | 0.9208 | 0.7998 |

Table S13: Cell-type specific element classification results (K562 vs. rest)

| Setting | Model | Accuracy | AUROC | AUPRC |
|---|---|---|---|---|
| Probed | Caduceus | 0.8533 ± 1.4897e-03 | 0.5873 | 0.1940 |
| | DNABERT-2 | 0.8550 ± 1.4821e-03 | 0.6913 | 0.2951 |
| | GENA-LM | 0.8563 ± 1.4766e-03 | 0.6929 | 0.3004 |
| | HyenaDNA | 0.8573 ± 1.4726e-03 | 0.7991 | 0.4390 |
| | Mistral-DNA | 0.8560 ± 1.4782e-03 | 0.6456 | 0.2589 |
| | Nucleotide Transformer | 0.8473 ± 1.5144e-03 | 0.7109 | 0.3209 |
| Fine-Tuned | Caduceus | 0.8391 ± 1.5469e-03 | **0.8776** | **0.5974** |
| | DNABERT-2 | 0.8358 ± 1.5595e-03 | 0.8715 | 0.5757 |
| | GENA-LM | 0.8361 ± 1.5585e-03 | 0.8622 | 0.5706 |
| | HyenaDNA | 0.8384 ± 1.5495e-03 | 0.8468 | 0.5333 |
| | Mistral-DNA | 0.8316 ± 1.5754e-03 | 0.7100 | 0.3239 |
| | Nucleotide Transformer | 0.8354 ± 1.5613e-03 | 0.8667 | 0.5707 |
| *Ab initio* | Probing-head-like | 0.8492 ± 1.5067e-03 | 0.7411 | 0.3343 |
| | ChromBPNet-like | **0.8645 ± 1.4408e-03** | 0.8475 | 0.4982 |

Table S14: Chromatin Accessibility Prediction Results (GM12878)

| Setting | Model | Spearman $r$ Peaks | Pearson $r$ Peaks | Spearman $r$ All | Pearson $r$ All | AUROC | AUPRC |
|---------|-------|--------------------|--------------------|------------------|------------------|-------|-------|
| Probed | Caduceus | 0.2510 | 0.3028 | 0.1751 | 0.2157 | 0.6053 | 0.4521 |
| | DNABERT-2 | 0.3946 | 0.4625 | 0.4899 | 0.5308 | 0.7570 | 0.6399 |
| | GENA-LM | 0.4899 | 0.5369 | 0.5014 | 0.5572 | 0.7836 | 0.6794 |
| | HyenaDNA | 0.3619 | 0.4125 | 0.3964 | 0.4693 | 0.7082 | 0.5707 |
| | Mistral-DNA | 0.2932 | 0.2669 | 0.2266 | 0.3418 | 0.5858 | 0.3692 |
| | Nucleotide Transformer | 0.4098 | 0.4556 | 0.4780 | 0.5191 | 0.7565 | 0.6271 |
| Fine-Tuned | Caduceus | 0.5029 | 0.5596 | 0.7405 | 0.7304 | 0.9350 | 0.8724 |
| | DNABERT-2 | 0.4892 | 0.5436 | 0.7304 | 0.7286 | 0.9157 | 0.8425 |
| | GENA-LM | 0.4669 | 0.5347 | 0.7196 | 0.7206 | 0.9084 | 0.8333 |
| | HyenaDNA | 0.4356 | 0.4962 | 0.6058 | 0.6069 | 0.8532 | 0.7452 |
| | Mistral-DNA | 0.3718 | 0.4368 | 0.5175 | 0.5557 | 0.7888 | 0.6694 |
| | Nucleotide Transformer | 0.5148 | 0.5862 | **0.7659** | **0.7650** | 0.9381 | **0.8868** |
| *Ab initio* | ChromBPNet | **0.5401** | **0.6074** | 0.7349 | 0.7282 | **0.9399** | 0.8851 |

Table S15: Chromatin Accessibility Prediction Results (H1ESC)

| Setting | Model | Spearman $r$ Peaks | Pearson $r$ Peaks | Spearman $r$ All | Pearson $r$ All | AUROC | AUPRC |
|---------|-------|--------------------|--------------------|------------------|------------------|-------|-------|
| Probed | Caduceus | 0.3706 | 0.4624 | 0.2484 | 0.3267 | 0.6076 | 0.4291 |
| | DNABERT-2 | 0.5835 | 0.6477 | 0.5714 | 0.6035 | 0.7629 | 0.6412 |
| | GENA-LM | 0.6779 | 0.7074 | 0.6311 | 0.6682 | 0.8093 | 0.7036 |
| | HyenaDNA | 0.5381 | 0.6070 | 0.5154 | 0.5630 | 0.7282 | 0.5932 |
| | Mistral-DNA | 0.4997 | 0.5002 | 0.4370 | 0.4700 | 0.6445 | 0.4424 |
| | Nucleotide Transformer | 0.5945 | 0.6544 | 0.5418 | 0.5611 | 0.7654 | 0.6504 |
| Fine-Tuned | Caduceus | 0.7437 | 0.7869 | 0.7983 | 0.7992 | 0.9541 | 0.9081 |
| | DNABERT-2 | 0.7173 | 0.7732 | 0.7810 | 0.7962 | 0.9405 | 0.8908 |
| | GENA-LM | 0.6962 | 0.7550 | 0.7768 | 0.7962 | 0.9416 | 0.8956 |
| | HyenaDNA | 0.6726 | 0.7271 | 0.7400 | 0.7486 | 0.9272 | 0.8610 |
| | Mistral-DNA | 0.5734 | 0.6478 | 0.6362 | 0.6835 | 0.8385 | 0.7353 |
| | Nucleotide Transformer | 0.7366 | 0.7969 | **0.8011** | **0.8150** | **0.9584** | **0.9247** |
| *Ab initio* | ChromBPNet | **0.7549** | **0.7971** | 0.7716 | 0.7534 | 0.9524 | 0.9062 |

Table S16: Chromatin Accessibility Prediction Results (HEPG2)

| Setting | Model | Spearman $r$ Peaks | Pearson $r$ Peaks | Spearman $r$ All | Pearson $r$ All | AUROC | AUPRC |
|---------|-------|--------------------|--------------------|------------------|------------------|-------|-------|
| Probed | Caduceus | 0.3123 | 0.3857 | 0.2623 | 0.3407 | 0.6108 | 0.5432 |
| | DNABERT-2 | 0.3566 | 0.4241 | 0.3342 | 0.3954 | 0.6499 | 0.5736 |
| | GENA-LM | 0.4008 | 0.4833 | 0.5052 | 0.5558 | 0.7709 | 0.7000 |
| | HyenaDNA | 0.3453 | 0.3962 | 0.3465 | 0.4072 | 0.6414 | 0.5506 |
| | Mistral-DNA | 0.3487 | 0.4096 | 0.3529 | 0.4141 | 0.6528 | 0.5586 |
| | Nucleotide Transformer | 0.3365 | 0.3989 | 0.3175 | 0.3862 | 0.6483 | 0.5777 |
| Fine-Tuned | Caduceus | 0.4536 | 0.5234 | 0.6671 | 0.6323 | 0.8964 | 0.8219 |
| | DNABERT-2 | 0.4719 | 0.5365 | 0.6858 | 0.6559 | 0.8934 | 0.8247 |
| | GENA-LM | 0.4392 | 0.5145 | 0.6626 | 0.6408 | 0.8777 | 0.8097 |
| | HyenaDNA | 0.4057 | 0.4782 | 0.6197 | 0.5949 | 0.8537 | 0.7732 |
| | Mistral-DNA | 0.3597 | 0.4241 | 0.4754 | 0.4833 | 0.7306 | 0.6331 |
| | Nucleotide Transformer | 0.5134 | 0.5773 | **0.7184** | **0.6876** | **0.9216** | **0.8690** |
| *Ab initio* | ChromBPNet | **0.5344** | **0.6021** | 0.6898 | 0.6711 | 0.9097 | 0.8618 |

Table S17: Chromatin Accessibility Prediction Results (IMR90)

| Setting | Model | Spearman $r$ Peaks | Pearson $r$ Peaks | Spearman $r$ All | Pearson $r$ All | AUROC | AUPRC |
|---------|-------|--------------------|--------------------|------------------|------------------|-------|-------|
| Probed | Caduceus | 0.1486 | 0.1719 | 0.2123 | 0.2163 | 0.6096 | 0.2702 |
| | DNABERT-2 | 0.2745 | 0.2884 | 0.4454 | 0.4448 | 0.7285 | 0.4125 |
| | GENA-LM | 0.3288 | 0.3638 | 0.5213 | 0.5539 | 0.7989 | 0.5410 |
| | HyenaDNA | 0.2371 | 0.2869 | 0.3856 | 0.4408 | 0.7016 | 0.3889 |
| | Mistral-DNA | 0.2439 | 0.2828 | 0.3866 | 0.3956 | 0.7116 | 0.4025 |
| | Nucleotide Transformer | 0.2693 | 0.3184 | 0.4379 | 0.4427 | 0.7387 | 0.4795 |
| Fine-Tuned | Caduceus | 0.4793 | 0.5258 | 0.7988 | 0.7475 | **0.9760** | 0.8997 |
| | DNABERT-2 | 0.4699 | 0.5126 | 0.7960 | 0.7505 | 0.9629 | 0.8580 |
| | GENA-LM | 0.4211 | 0.4778 | 0.7898 | 0.7512 | 0.9612 | 0.8569 |
| | HyenaDNA | 0.4255 | 0.4703 | 0.7231 | 0.6737 | 0.9412 | 0.7851 |
| | Mistral-DNA | 0.3023 | 0.3425 | 0.5912 | 0.5830 | 0.8547 | 0.6093 |
| | Nucleotide Transformer | 0.4890 | 0.5416 | **0.8195** | **0.7806** | 0.9745 | **0.9018** |
| *Ab initio* | ChromBPNet | **0.5495** | **0.5963** | 0.7749 | 0.7314 | 0.9745 | 0.8886 |

Table S18: Chromatin Accessibility Prediction Results (K562)

| Setting | Model | Spearman $r$ Peaks | Pearson $r$ Peaks | Spearman $r$ All | Pearson $r$ All | AUROC | AUPRC |
|---------|-------|------------------|-----------------|----------------|---------------|-------|-------|
| Probed | Caduceus | 0.4006 | 0.5499 | 0.3009 | 0.4444 | 0.6164 | 0.5819 |
| | DNABERT-2 | 0.4827 | 0.6215 | 0.4811 | 0.5722 | 0.7208 | 0.6701 |
| | GENA-LM | 0.4610 | 0.6152 | 0.4988 | 0.5811 | 0.7607 | 0.7066 |
| | HyenaDNA | 0.4381 | 0.5616 | 0.3763 | 0.4624 | 0.6621 | 0.5869 |
| | Mistral-DNA | 0.4307 | 0.5634 | 0.3973 | 0.5019 | 0.6782 | 0.6031 |
| | Nucleotide Transformer | 0.4990 | 0.6339 | 0.5116 | 0.6037 | 0.7640 | 0.7203 |
| Fine-Tuned | Caduceus | 0.5698 | 0.6668 | 0.7599 | 0.7475 | 0.9334 | 0.8852 |
| | DNABERT-2 | 0.5286 | 0.6484 | 0.7357 | 0.7329 | 0.9172 | 0.8674 |
| | GENA-LM | 0.5323 | 0.6392 | 0.7349 | 0.7389 | 0.9096 | 0.8603 |
| | HyenaDNA | 0.4456 | 0.5878 | 0.5112 | 0.5693 | 0.7499 | 0.6729 |
| | Mistral-DNA | 0.4305 | 0.5678 | 0.5615 | 0.6005 | 0.7956 | 0.7117 |
| | Nucleotide Transformer | **0.5829** | **0.6863** | **0.7764** | **0.7714** | **0.9412** | **0.9016** |
| *Ab initio* | ChromBPNet | 0.5741 | 0.6687 | 0.7200 | 0.7246 | 0.9167 | 0.8762 |

**African caQTLs (Supervised)**

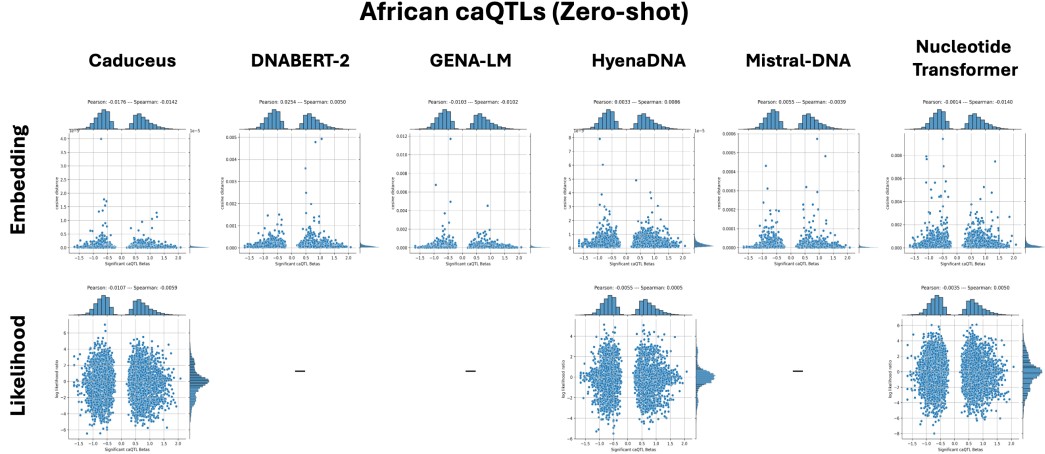

Figure S6: African LCLs caQTLs Supervised Model Scores

**African caQTLs (Zero-shot)**

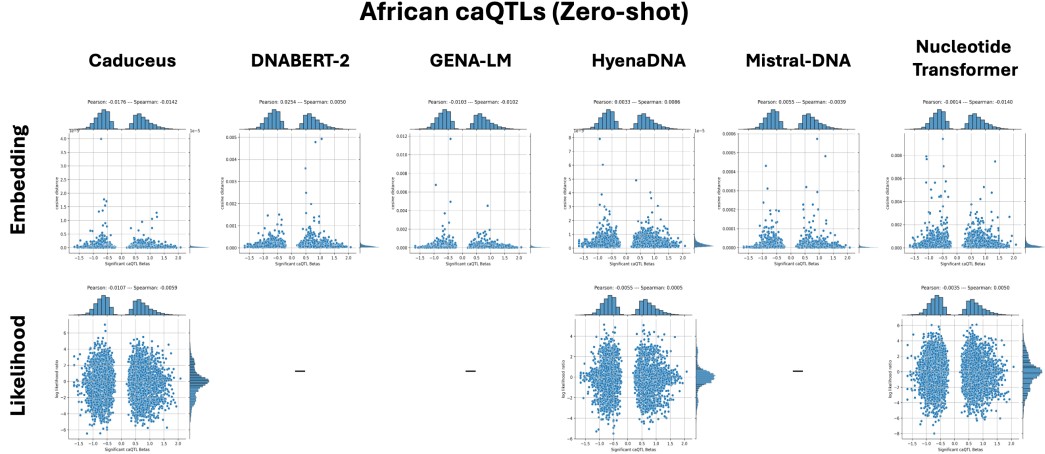

Figure S7: African LCLs caQTLs Zero Shot Model Scores

**Yoruban dsQTLs (Supervised)**

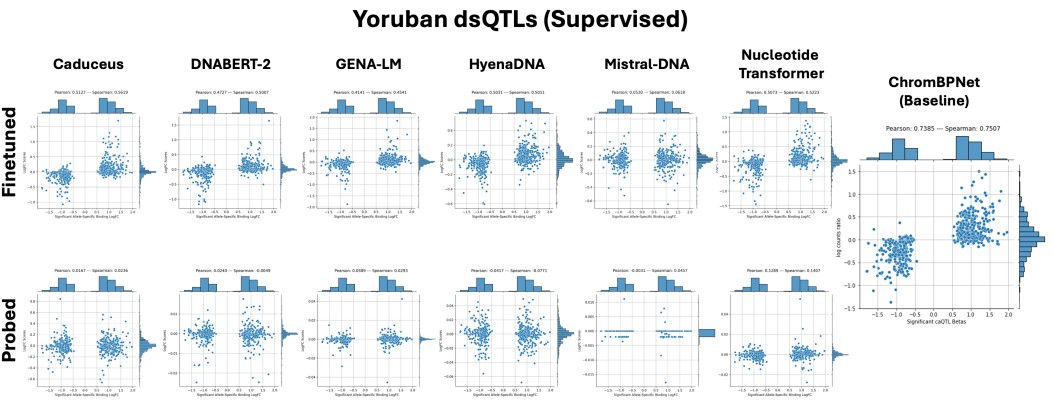

Figure S8: Yoruban LCLs dsQTLs Supervised Model Scores

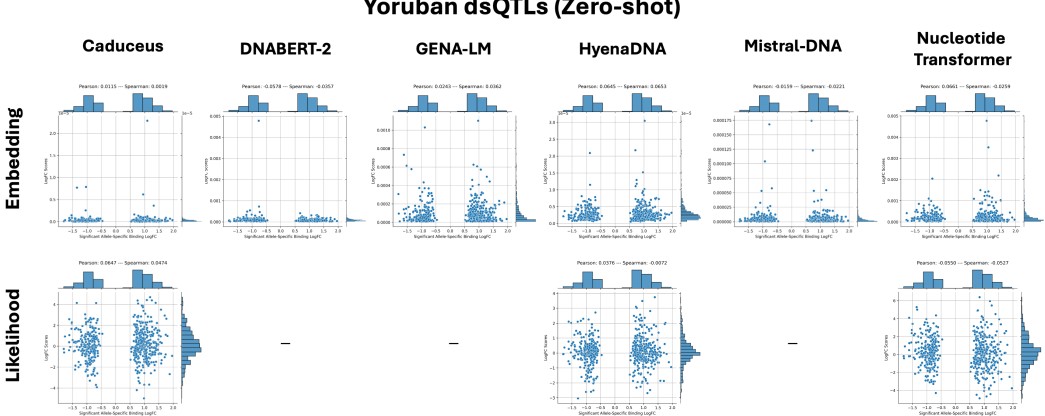

Figure S9: Yoruban LCLs dsQTLs Zero Shot Model Scores

Table S19: African caQTL Supervised Variant Scoring Extended Results

| Setting | Model | Pearson $r$ | Spearman $r$ | AUROC | AUPRC | Wilcoxon $p$-value |
|---|---|---|---|---|---|---|
| Probed | Caduceus | -0.0044 | 0.0040 | 0.5124 | 0.0848 | 0.0003 |
| | DNABERT-2 | 0.0064 | 0.0111 | 0.5024 | 0.0810 | 0.2536 |
| | GENA-LM | -0.0067 | 0.0082 | 0.5149 | 0.0836 | 2.2893e-5 |
| | HyenaDNA | 0.0121 | 0.0139 | 0.5658 | 0.0937 | 4.0326e-73 |
| | Mistral-DNA | 0.0185 | 0.0142 | 0.5018 | 0.0821 | 0.2706 |
| | Nucleotide Transformer | 0.0058 | 0.0052 | 0.5248 | 0.0870 | 5.4018e-12 |
| Fine-Tuned | Caduceus | 0.2591 | 0.2818 | 0.6498 | 0.1791 | **0.000** |
| | DNABERT-2 | 0.1840 | 0.2185 | 0.6155 | 0.1380 | 1.1660e-220 |
| | GENA-LM | 0.2011 | 0.2161 | 0.6038 | 0.1285 | 1.2926e-178 |
| | HyenaDNA | 0.2653 | 0.2871 | 0.6108 | 0.1233 | 4.358e-203 |
| | Mistral-DNA | 0.0849 | 0.0858 | 0.5101 | 0.0841 | 0.0027 |
| | Nucleotide Transformer | 0.2299 | 0.2435 | 0.6231 | 0.1542 | 2.8400e-250 |
| *Ab initio* | ChromBPNet | **0.6712** | **0.6995** | **0.7716** | **0.3972** | **0.000** |

Table S20: Yoruban dsQTL Supervised Variant Scoring Extended Results

| Setting | Model | Pearson $r$ | Spearman $r$ | AUROC | AUPRC | Wilcoxon $p$-value |
|---|---|---|---|---|---|---|
| Probed | Caduceus | 0.0167 | 0.0236 | 0.4901 | 0.0200 | 0.7893 |
| | DNABERT-2 | 0.0240 | -0.0049 | 0.4756 | 0.0194 | 0.9760 |
| | GENA-LM | 0.0589 | 0.0292 | 0.4655 | 0.0191 | 0.9975 |
| | HyenaDNA | -0.0417 | -0.0771 | 0.4672 | 0.0187 | 0.9961 |
| | Mistral-DNA | -0.0031 | 0.0457 | 0.4324 | 0.0201 | 1.000 |
| | Nucleotide Transformer | 0.1289 | 0.1407 | 0.5163 | 0.0222 | 0.0934 |
| Fine-Tuned | Caduceus | 0.5127 | 0.5619 | 0.6664 | 0.0764 | 8.176e-42 |
| | DNABERT-2 | 0.4727 | 0.5007 | 0.6307 | 0.0416 | 1.390e-26 |
| | GENA-LM | 0.4141 | 0.4541 | 0.6280 | 0.0396 | 1.524e-25 |
| | HyenaDNA | 0.5031 | 0.5011 | 0.5729 | 0.0289 | 1.6439e-09 |
| | Mistral-DNA | 0.0530 | 0.0618 | 0.5041 | 0.0204 | 0.3697 |
| | Nucleotide Transformer | 0.5073 | 0.5223 | 0.6697 | 0.0796 | 1.949e-43 |
| *Ab initio* | ChromBPNet | **0.7385** | **0.7507** | **0.8916** | **0.3587** | **7.610e-222** |

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
