# OpenReview forum: "DART-Eval: A Comprehensive DNA Language Model Evaluation Benchmark on Regulatory DNA"
_NeurIPS.cc/2024/Datasets_and_Benchmarks_Track — NeurIPS 2024 Track Datasets and Benchmarks Poster_

### Official Review · Reviewer_jn6m · 2024-07-23

**Rating:** 6
**Confidence:** 5
**Correctness:** The claims made in the submission are…
**Clarity:** Please refer to "Review" for my comme…

**Review:**

## 1. Quality:

The overall quality of this work is high. The authors have conducted a thorough and well-designed study that addresses a significant gap in the field of DNA language models (DNALMs) for regulatory genomics.

### 1.1 Pros:

- **(P1) Rigorous methodology:** The benchmark suite (DART-Eval) is carefully designed, incorporating multiple tasks of varying complexity and addressing key biological confounders.

- **(P2) Thorough evaluation:** The study examines multiple state-of-the-art DNALMs across different settings (zero-shot, probed, and fine-tuned) and compares them against strong baseline models.

- **(P3) High-quality datasets:** The benchmark utilizes reputable data sources such as ENCODE and HOCOMOCO, ensuring the reliability of the evaluation.

- **(P4) Careful consideration of biological context:** The authors demonstrate a deep understanding of regulatory genomics, which is reflected in their task design and analysis.

### 1.2 Cons:

- **(C1) Lack of in-depth analysis:** The paper would benefit from more in-depth analysis of how different model components contribute to performance.

- **(C2) Limited exploration of architectural variations:** While multiple models are evaluated, there's little investigation into how specific architectural choices impact performance on regulatory genomics tasks.

- **(C3) Insufficient discussion and analysis of recent work:** Recent state-of-the-art DNALMs like VQDNA[1] has been overlooked.

## Reference

**[1]** VQDNA: Unleashing the Power of Vector Quantization for Multi-Species Genomic Sequence Modeling, ICML 2024;

---

## 2. Clarity:

The paper is generally well-written and organized, making it accessible to researchers in both machine learning and genomics.

### 2.1 Pros:

- **(P1) Clear structure:** The paper follows a logical flow, from motivation to methodology to results and discussion.

- **(P2) Effective illustration of complex concepts:** The paper greatly explain intricate biological concepts and their relevance to machine learning models.

- **(P3) Detailed benchmarking methodology:** The experimental setup and evaluation metrics are well-described, enhancing reproducibility.

### 2.2 Cons:

- **(C1) Dense information:** Some sections, particularly those describing the benchmark tasks, could benefit from additional visual aids to improve clarity.

---

## 3. Originality:

The work presents a novel and valuable contribution to the field.

### 3.1 Pros:

- **(P1) Novel benchmark:** DART-Eval is the first comprehensive benchmark specifically designed for evaluating DNALMs on regulatory genomics tasks.

- **(P2) Unique focus:** The paper addresses an important but understudied area at the intersection of deep learning and regulatory genomics.

- **(P3) Innovative task design:** The benchmark incorporates tasks that test various aspects of regulatory genomics, from basic syntax to complex, context-dependent predictions.

### 3.2 Cons:

- **(C1) Limited novelty in methods:** While the evaluation framework is novel, the paper doesn't introduce new DNA modeling methods.

---

## 4. Significance:

The significance of this work is substantial for both the machine learning and genomics communities.

### 4.1 Pros:

- **(P1) Addressing a critical gap:** The paper provides a much-needed evaluation for DNALMs in regulatory genomics tasks.
- **(P2) Impactful findings:** The observation that existing DNALMs don't consistently outperform specialized models has important implications for future research directions.
- **(P3) Practical relevance:** The benchmark and findings provide valuable guidance for researchers and practitioners working on applied genomics problems.
- **(P4) Potential for broad impact:** The work could influence the development of more effective DNALMs for regulatory genomics, potentially advancing our understanding of gene regulation.

### 4.2 Cons:

- **(C1) Limited exploration of practical applications:** While the potential impact is high, the paper doesn't extensively discuss how the findings might contribute to specific applied genomics problems.

**Strengths:**

**(S1)** This work provides a thorough evaluation framework for DNALMs, addressing a critical gap in the AI for Genomics community. The DART-Eval benchmark covers a wide range of biologically relevant tasks, considering confounding factors and using high-quality datasets from reputable sources like ENCODE and HOCOMOCO.

**(S2)** The evaluation consists of multiple recent DNALMs across different settings (zero-shot, probed, and fine-tuned) and compares them against strong baselines. This comprehensive evaluation provides a clear picture of the current state of DNALMs in regulatory genomics tasks.

**(S3)** This paper, unlike others, shows a deep understanding of the remaining challenges in regulatory genomics, addressing issues like G/C content bias, linkage disequilibrium, and cell-type specificity. This attention to detail enhances the reliability and biological relevance of the results, offering valuable insights for potential improvement in model architecture, training strategies, and data curation.

**Additional Feedback:**

I hope my review helps to further strengthen this work and helps the authors, fellow reviewers, and Area Chairs understand the basis for my recommendation. I look forward to the rebuttal feedback and would be glad to raise my rating if thoughtful responses and improvements are presented.

**Documentation:**

There is sufficient detail on data collection and organization, availability and maintenance, and ethical and responsible use.

**Ethics:**

There are no or only very minor ethical concerns with the submission that warrant further discussion or review.

**Limitations:**

The authors have adequately addressed the limitations and potential negative societal impact of their work.

**Opportunities For Improvement:**

## Limitations:

**(L1) Limited exploration of model architectures:**
While the paper evaluates several DNALMs, it does not explore the impact of different model architectures (e.g., local attention, hierarchical attention) which might affect performance on tasks requiring understanding of both local and long-range genomic context. Such exploration could provide more insights into designing better DNALMs for regulatory genomics tasks.

**(L2) Lack of ablation studies:**
The paper would benefit from ablation studies to isolate the effects of different components (e.g., pre-training objectives, and tokenization strategies) on model performance across tasks. Such studies could also determine the impact of different tokenization strategies (e.g., single-nucleotide vs. k-mer vs. byte-pair encoding vs. VQ tokenization [1]) on model performance and computational efficiency.

**(L3) Insufficient discussion and analysis of recent work:** Recent state-of-the-art DNALMs like VQDNA [1] has been overlooked.

**(L4) Insufficient discussion of computational efficiency:**
Although the paper mentions that DNALMs require more computational resources, a more detailed analysis of the trade-offs between model size, performance, and computational requirements would be valuable for practical applications. This could include quantitative comparisons of training/inference time and memory footprint for different models across tasks.

---

## Reference

**[1]** VQDNA: Unleashing the Power of Vector Quantization for Multi-Species Genomic Sequence Modeling, ICML 2024;

**Relation To Prior Work:**

This work has clearly discussed how it differs from previous contributions.

**Summary And Contributions:**

This paper introduces the DART-Eval benchmark for evaluating DNA language models (DNALMs) on biologically significant regulatory genomics tasks. It comprises five types of tasks with increasing difficulty, such as functional sequence feature discovery, cell-type specific regulatory activity prediction, and genetic variant impact prediction. It evaluates several recent DNALMs in zero-shot, probed, and fine-tuned settings, comparing them against state-of-the-art ab initio models. The results demonstrate that existing DNALMs fail to show consistent performance and compelling advantages over baselines for most tasks while requiring more computational costs. The benchmark contributes valuable insights into the limitations of DNALMs and proposes strategies for improving their performance in regulatory genomics applications.

---

> ### Author Rebuttal · Authors · 2024-08-16
>
> We appreciate the detailed and thoughtful review. Below, we address each point in turn.
>
> # Quality
>
> **(C1) Lack of in-depth analysis: The paper would benefit from more in-depth analysis of how different model components contribute to performance.**
>
> We thank the reviewer for the suggestion for a more in-depth analysis of model components, which if we understand correctly, concerns tokenization and model objectives (e.g. masked vs. autoregressive).
>
> In terms of performance metrics, we do not find systematic differences among modeling aspects. Nevertheless, these modeling decisions heavily influence the computational feasibility of our evaluations, which we briefly touch upon in the paper. We will make sure to include a more detailed discussion of these aspects in our manuscript.
>
> **(C2) Limited exploration of architectural variations: While multiple models are evaluated, there's little investigation into how specific architectural choices impact performance on regulatory genomics tasks.**
>
> We agree that comparisons between different DNALM architectures would be scientifically useful. However, we do not find conclusive differences between unsupervised architectures in our evaluations. Hence, we focus on performance aspects shared across all tested DNALMs stemming from pre-training methodology. We elaborate on these points further in the global rebuttal.
>
> **(C3) Insufficient discussion and analysis of recent work: Recent state-of-the-art DNALMs like VQDNA[1] has been overlooked.**
>
> We thank the reviewer for bringing this model to our attention. We will be sure to cite VQDNA. Additionally, we are happy to evaluate VQDNA but we are unable to locate the model code on GitHub or Huggingface. We have also begun evaluating additional models and have included initial results in the attachment.
>
> # Clarity
>
> **(C1) Dense information: Some sections, particularly those describing the benchmark tasks, could benefit from additional visual aids to improve clarity.**
>
> We thank the reviewer for this suggestion. We will add additional figures and visualizations to help readers understand the evaluations and conclusions.
>
> # Originality
>
> **(C1) Limited novelty in methods: While the evaluation framework is novel, the paper doesn't introduce new DNA modeling methods.**
>
> As a Datasets and Benchmarks track paper, we intend the main contribution to be well-motivated benchmark datasets, evaluation tasks, and metrics for DNALMs with a specific focus on regulatory DNA. We hope that these benchmarks will help guide and validate future DNA modeling methods.
>
> # Significance
>
> **(C1) Limited exploration of practical applications: While the potential impact is high, the paper doesn't extensively discuss how the findings might contribute to specific applied genomics problems.**
>
> We thank the reviewer for bringing up this important point. We hope to contribute to downstream biological problems by allowing researchers to design more effective DNALMs. As such, our framework focuses on evaluating DNALMs for the most important downstream applications of predictive models of regulatory DNA, including the interpretation of sequence features and syntax, counterfactual predictions of variant effects, and design of synthetic sequences. Previously published benchmark datasets and tasks used for evaluating regulatory DNALMs have not focused on these critical applications. We will include these clarifications in the manuscript.
>
> # Limitations (overall)
>
> **(L1) Limited exploration of model architectures: While the paper evaluates several DNALMs, it does not explore the impact of different model architectures (e.g., local attention, hierarchical attention) which might affect performance on tasks requiring understanding of both local and long-range genomic context. Such exploration could provide more insights into designing better DNALMs for regulatory genomics tasks.**
>
> Please see responses to Quality: (C1, C2)
>
> **(L2) Lack of ablation studies: The paper would benefit from ablation studies to isolate the effects of different components (e.g., pre-training objectives, and tokenization strategies) on model performance across tasks. Such studies could also determine the impact of different tokenization strategies (e.g., single-nucleotide vs. k-mer vs. byte-pair encoding vs. VQ tokenization [1]) on model performance and computational efficiency.**
>
> As a Datasets and Benchmarks submission, we intend the primary contribution of this study to be a rigorous set of evaluations for DNA language models. A thorough ablation study and architecture search, though immensely interesting, would fall beyond the intended scope of our work.
>
> That being said, we do find that objectives and tokenization strategies heavily influence computational efficiency at a purely asymptotic level, which we briefly touch upon in the manuscript. We agree that runtime is an important practical concern and we will expand our discussion of this topic.
>
> **(L3) Insufficient discussion and analysis of recent work: Recent state-of-the-art DNALMs like VQDNA [1] has been overlooked.**
>
> See response to Quality: (C3)
>
> **(L4) Insufficient discussion of computational efficiency: Although the paper mentions that DNALMs require more computational resources, a more detailed analysis of the trade-offs between model size, performance, and computational requirements would be valuable for practical applications. This could include quantitative comparisons of training/inference time and memory footprint for different models across tasks.**
>
> We thank the reviewers for raising this important point. We provide information regarding the hardware used for each part of our study. Evaluations on all models require no more than a single Nvidia A100 GPU. We summarized resource requirements in the attached pdf.
>
> We furthermore briefly touch on the impact of different tokenization and objectives on runtime. We will add a more in-depth discussion of factors affecting performance in our manuscript.

---

### Official Review · Reviewer_xoes · 2024-07-25

**Rating:** 6
**Confidence:** 3
**Correctness:** Seems correct
**Clarity:** Overall well written

**Review:**

Please see Strengths and Opportunities For Improvement.

**Strengths:**

1. Addresses an important gap in evaluating DNALMs on regulatory DNA tasks

2. Comprehensive benchmark covering diverse and biologically relevant tasks and careful design of datasets to control for biological confounders

3. Rigorous evaluation across zero-shot, probed, and fine-tuned settings and insightful analysis of DNALM components like tokenization and positional encoding

**Additional Feedback:**

Please see Opportunities For Improvement.

**Documentation:**

Documentation not provided for most of the code and should be improved

**Limitations:**

The paper includes discussions on limitations.

**Opportunities For Improvement:**

1. No comparison to recent DNALMs like Caduceus or Mistral-DNA. Including discussions on them will add to comprehensiveness.

2. The code lacks documentation, which is a necessary part.

3. Limited exploration of model architectures. The paper doesn't deeply explore how different architectural choices in DNALMs impact performance on various tasks.

4. Lack of human baseline. For some tasks, it might be informative to include a human expert baseline for comparison.

5. Limited discussion of computational resources required for each model, which is important for reproducibility and practical considerations.

6. Limited discussion of statistical significance for model performances. Also limited error analysis.

**Relation To Prior Work:**

The paper includes comparisons with prior works

**Summary And Contributions:**

This paper introduces DART-Eval, a benchmark for evaluating DNA language models on regulatory DNA tasks. It includes 5 tasks covering regulatory element identification, transcription factor binding motif sensitivity, cell-type specific regulatory syntax, quantitative regulatory activity prediction, and variant effect prediction. The authors evaluate several state-of-the-art DNALMs in zero-shot, probed, and fine-tuned settings against ab initio baselines. Key findings include that DNALMs show promise on simpler tasks but struggle with more complex ones, embedding-based approaches underperform likelihood-based ones, and fine-tuned DNALMs often do not outperform ab initio models despite higher computational costs.

---

> ### Author Rebuttal · Authors · 2024-08-16
>
> We appreciate the detailed and thoughtful review. Below, we address each point in turn.
>
> **No comparison to recent DNALMs like Caduceus or Mistral-DNA. Including discussions on them will add to comprehensiveness.**
>
> We thank the reviewer for suggesting additional relevant DNALMs for comparison. We aim to add these additional comparisons to the manuscript. We have begun the process of evaluating additional models, and we have included some initial results in the attached pdf.
>
> **The code lacks documentation, which is a necessary part.**
>
> We thank the reviewer for pointing out the importance of documentation. We have added additional documentation since the submission deadline, with an emphasis on reproducibility and dataset provenance. We are actively working on more thoroughly documenting our code and making it more user-friendly.
>
> **Limited exploration of model architectures. The paper doesn't deeply explore how different architectural choices in DNALMs impact performance on various tasks.**
>
> We agree that comparisons between different DNALM architectures would be scientifically useful. Our study aims to evaluate existing DNALMs that are trained genome-wide without using any external annotation information. The evaluated DNALMs employ a diversity of model sizes, objectives, architectures, and context lengths. We do not find any conclusive differences in performance between specific architecture choices in our evaluations. However, a universally shared feature is that they are all trained on sequences randomly sampled across one or more entire genomes. However, this naive sampling strategy does not account for the heterogeneity of functional elements (e.g. coding DNA, promoters, enhancers, splice sites, 5’-UTRs, 3’-UTRs) each of which has a very different relative prevalence in the genome and encode distinct and diverse functional syntax. Furthermore, mammalian genomes are dominated by neutral DNA sequences with functional DNA elements accounting for a small fraction of the genome. Hence, we believe that the performance shortcomings of current genome-wide DNALMs primarily arise as a result of not using training data sampling strategies that would help overcome the challenges noted above, rather than from architecture or objectives. We believe that a truly effective DNALM will need to rectify these data sampling issues before optimizing for architecture.
>
> **Lack of human baseline. For some tasks, it might be informative to include a human expert baseline for comparison.**
>
> We agree that human baselines would be immensely useful. However, high-quality human baselines are unavailable for chromatin activity and variant-scoring tasks given the cell-type-specific and dataset-specific nature of these analyses. Comprehensive human annotation at an experiment-specific level would require vast resources beyond our capabilities.
>
> That being said, some of our evaluations test general sensitivity to functional syntax. For those, we leverage the reputable ENCODE cCRE and HOCOMOCO motif datasets, which are carefully curated by expert scientists.
>
> **Limited discussion of computational resources required for each model, which is important for reproducibility and practical considerations.**
>
> We thank the reviewers for raising this important point. We provide information regarding the hardware used for each part of our study. Evaluations on all models require no more than a single Nvidia A100 GPU. We have summarized the resource requirements of the evaluated DNALMs in the attached pdf.
>
> We furthermore briefly touch on the impact of different tokenization and objectives on runtime. We will add a more in-depth discussion of factors affecting performance in our manuscript.
>
> **Limited discussion of statistical significance for model performances. Also limited error analysis.**
>
> We agree with this point about statistical significance and error analysis. We will add confidence intervals when appropriate (e.g. using the binomial test for accuracy metrics). We anticipate that intervals will be quite tight given the large number of test samples for each evaluation.

---

> > ### Comment · Reviewer_xoes · 2024-08-29
> > **Response to Rebuttal**
> >
> > I have read the rebuttal and appreciate the authors' further explanations and added experiments. Several of my questions are well-addressed. I also look forward to the refined version of the paper.

---

### Official Review · Reviewer_UVbC · 2024-07-25
**Impressive manuscript**

**Rating:** 7
**Confidence:** 4
**Correctness:** The claims are correct.
**Clarity:** Yes

**Review:**

The DNA LMs are understudied and are difficult to compare. This benchmark aims to solve this problem. The experiments are defined in detail, along with their relevance. The LMs are compared against much simpler supervised methods. The authors also perform fine-tuning on DNA LMs for fairer comparison. The models are compared on a variety of datasets for each of the tasks ensuring robustness of the results and conclusions. It is also interesting to see that simpler supervised models often are at par with or outperform the DNA LMs. However, one might need to compare these to more challenging regression-based tasks like chromatin structure prediction.

**Strengths:**

DNA LMs are very powerful and exciting tools. However, their systematic comparison is needed, and this paper tries to tackle this exact problem.

**Additional Feedback:**

No additional feedback

**Documentation:**

Sufficient detail to support reproducibility is there.

**Ethics:**

No ethical concerns.

**Limitations:**

Limitations are mentioned by the authors.

**Opportunities For Improvement:**

Same as mentioned above.

**Relation To Prior Work:**

Yes

**Summary And Contributions:**

The authors propose a benchmark to study the applicability of DNA Language Models (LMs) to various regulatory downstream tasks. Four LMs are considered: DNABERT2, Nucleotide Transformer, GENA-LM, and HyenaDNA. The authors make important inferences, which are commented upon below.

---

> ### Author Rebuttal · Authors · 2024-08-16
>
> We greatly appreciate the review and the suggestions, and we address the main points below.
>
> **One might need to compare [the simpler supervised models] to more challenging regression-based tasks like chromatin structure prediction.**
>
> We agree, and state in our paper, that future directions for regulatory benchmarking include a variety of additional classes of sequences, including 3’ UTRs, 5’ UTRs, splice sites, and sequences affecting chromatin structure. Additionally, chromatin structure’s long-context nature offers unique modeling challenges, and we believe that rigorous evaluations of this property would constitute a substantial paper by itself. However, effectively learning local sequence syntax predictive of short-range regulatory activity (e.g. chromatin accessibility) is a prerequisite for any model that aims to effectively learn long-range regulatory or structural interactions. Our evaluations focus on local regulatory syntax, showing that current DNALMs do not effectively learn local syntax. Hence, we do not expect them to perform well on long-range regulatory and structure prediction tasks. We are keen to follow up on long-range chromatin structure prediction tasks once we see evidence that DNALMs can effectively learn and solve local prediction tasks.

---

### Official Review · Reviewer_jaVt · 2024-07-26
**The review of DART-Eval: A Comprehensive Benchmark for Evaluating DNA Language Models on Regulatory DNA**

**Rating:** 7
**Confidence:** 4

**Review:**

Quality: The quality of the research is commendable, addressing a critical gap in the evaluation of DNALMs by providing a comprehensive and standardized benchmark.

Clarity: The paper is well-written and clearly presents its objectives, methodologies, and findings.

Originality: The introduction of a benchmark specifically focused on regulatory DNA is a novel contribution to the field.

Significance: The work is significant for the broader research community, offering a much-needed tool for the evaluation and improvement of DNALMs.

**Strengths:**

1. Identifies limitations of existing regulatory DNA datasets and addresses them with a new benchmark.

2. Provides a standardized platform for evaluating DNALMs, which is essential for future advancements in the field.

**Additional Feedback:**

The paper makes a significant contribution to the field by introducing a novel benchmark for evaluating DNALMs. Addressing the noted limitations and expanding on certain analyses would further strengthen the work.

**Clarity:**

The paper is well-written and clearly presents its objectives, methodologies, and findings.

**Correctness:**

The claims made in the submission are generally correct, and the benchmark is constructed in a sound way. The evaluation methods and experiment design are appropriate and performed correctly.

**Documentation:**

There is sufficient detail on data collection and organization, availability and maintenance, and ethical and responsible use. The documentation supports reproducibility.

**Ethics:**

There are no significant ethical concerns with the submission.

**Limitations:**

The authors have acknowledged the limitations of their work but could further address how linkage disequilibrium (LD) is managed within the dataset. Additionally, testing motif correlation in models without pre-training and expanding the analysis of transcription factor binding motifs would enhance the robustness of their findings.

**Opportunities For Improvement:**

- Provide a clearer explanation of how linkage disequilibrium (LD) is addressed within the dataset.
- Test motif correlation in a model that has not undergone pre-training to strengthen the claim about the pre-training data distribution.
- Expand the analysis of transcription factor binding motifs in Chapter 4.2 to explore factors contributing to accuracy differences, such as the impact of insertion sites or motif types.

**Relation To Prior Work:**

The paper clearly discusses how this work differs from previous contributions, particularly in addressing the limitations of existing regulatory DNA datasets and providing a standardized evaluation platform for DNALMs.

**Summary And Contributions:**

DART-Eval introduces a novel benchmark to rigorously assess the performance of DNA language models (DNALMs) in the context of regulatory DNA. The paper compares current DNALMs with state-of-the-art ab initio models across a range of regulatory DNA tasks, revealing inconsistencies in the performance of existing models. DART-Eval aims to provide a standardized platform for evaluating and improving DNALMs, addressing limitations in existing regulatory DNA datasets with a meticulously curated benchmark dataset that enhances the scope of recent studies.

---

> ### Author Rebuttal · Authors · 2024-08-16
>
> We appreciate the detailed and thoughtful review. Below, we address each point in turn.
>
> **Provide a clearer explanation of how linkage disequilibrium (LD) is addressed within the dataset.**
>
> In our variant evaluation benchmark, we are restricting to QTLs associated with changes in chromatin accessibility measured by DNase-seq and ATAC-seq experiments. Linkage disequilibrium (LD) is a problem when identifying causal variants from QTL studies. However, we further restrict our “positive set” of likely causal variants to local cis-QTLs of DNase/ATAC-seq peaks with the additional restriction that the variant be located within its associated peak. The rationale for this decision is that causal variants within a locus affecting the accessibility of a peak region are most likely to disrupt transcription factor binding sites within the peak itself. Hence, by restricting to cis-QTL variants inside the associated peaks, we strongly enrich for likely causal variants. This filtering furthermore excludes all distal QTLs (causal variants in one peak that impact the accessibility of a distal peak via long-range interactions). Instead, we focus on likely causal cis-acting local variants. We thank the reviewer for bringing up this point and we will make the mitigating factors more clear in our manuscript.
>
> **Test motif correlation in a model that has not undergone pre-training to strengthen the claim about the pre-training data distribution.**
>
> If we understand correctly, the reviewer proposes to run the motif sensitivity evaluations (Chapter 4.2) on un-initialized DNALMs. We agree that this analysis is a useful null comparison, and we thank the reviewer for this idea. Results are included in the attached PDF.
>
> **Expand the analysis of transcription factor binding motifs in Chapter 4.2 to explore factors contributing to accuracy differences, such as the impact of insertion sites or motif types.**
>
> We thank the reviewer for these helpful comments. We do explore specific families of transcription factors, which we mention in the text (Figure S3). We find substantial variation in sensitivity across different families of TFs. We will elaborate more on this point in the manuscript.
>
> Regarding insertion sites, we agree that the background context can substantially impact motif syntax. To mitigate such biases, our evaluations utilize simple dinucleotide-shuffled synthetic backgrounds. A deep dive into specific endogenous backgrounds would be of scientific interest but would be a major separate undertaking beyond the scope of what we intend to show with this simple test for motif sensitivity.

---

### Author Rebuttal · Authors · 2024-08-16

Thank you all for the detailed feedback on our manuscript. We appreciate the reviewers’ thorough evaluations and constructive suggestions. Below, we address the key points raised.

The primary goal of our paper, as a Datasets and Benchmarks Track submission, is to introduce a set of rigorous, biologically motivated evaluations for DNA language models specifically focused on regulatory DNA. As such, we focus on benchmarking existing DNALMs and leave exploratory architecture searches to future model development work.

A key finding of our study is that existing DNALMs generally display unstable performance, lacking desired inductive biases in zero-shot tasks and failing to compellingly outperform ab initio models in fine-tuned supervised tasks. The evaluated DNALMs employ a diversity of model sizes, objectives, architectures, and context lengths, but are all trained on sequences randomly sampled across entire genomes. However, this naive sampling strategy does not account for the heterogeneity of functional elements (e.g. coding DNA, promoters, enhancers, splice sites, 5’-UTRs, 3’-UTRs), with each class having a very different relative prevalence in the genome and encoding distinct and diverse functional syntax. Further, mammalian genomes are dominated by neutral DNA sequences with functional DNA elements accounting for a small fraction of the genome. Hence, we believe that the performance shortcomings of current genome-wide DNALMs primarily arise as a result of not using training data sampling strategies that would help overcome the challenges noted above, rather than from architecture or objectives. A truly effective DNALM would need to rectify these data sampling issues before optimizing for architecture.

Several reviewers requested more information on the resource requirements of the evaluated models. We find that certain modeling decisions significantly impact runtime, which we briefly touch upon in the manuscript. For example, calculating sequence likelihoods is more efficient for autoregressive rather than masked models. We also point out how byte-pair encoding tokenization hinders comparisons of similar sequences, relevant to a variety of important applications including variant effect interpretation. We have summarized the resource requirements of the DNALMs evaluated in this study in the attached pdf, and we will include a more thorough discussion in the revised manuscript.

Reviewers pointed out the omission of recent DNALMs such as Caduceus, Mistral-DNA, and VQDNA. We acknowledge this gap and agree that including additional models would make our study more comprehensive. We have begun the process of evaluating additional models and we have included some initial results in the attached pdf.

Once again, we are grateful for the constructive feedback from all reviewers, and we thank you for your time and effort in reviewing our submission. We look forward to incorporating your suggestions and to further discussions of our manuscript.

---

### Decision · Program_Chairs · 2024-09-26

**Decision:**

Accept (Poster)

**Comment:**

This paper introduces a new DNA language model benchmark focused on regulatory DNA such as transcription factor motif sensitivity. The paper finds that current DNA LMs are lacking compared to domain-specific ones.

All reviewers appreciate the contribution and agree that this potentially is a paper that adds to the field in a significant way. The paper is well-written and well-executed. It complement existing benchmarks and will therefore help maturing the field and have significant impact